# Exact, Tractable Gauss-Newton Optimization in Deep Reversible Architectures Reveal Poor Generalization

**Davide Buffelli**[*]
MediaTek Research

**Jamie McGowan**[*]
MediaTek Research

**Wangkun Xu**[†]
Imperial College London

**Alexandru Cioba**
MediaTek Research

**Da-shan Shiu**
MediaTek Research

**Guillaume Hennequin**
MediaTek Research & University of Cambridge

**Alberto Bernacchia**
MediaTek Research

## Abstract

Second-order optimization has been shown to accelerate the training of deep neural networks in many applications, often yielding faster progress per iteration on the training loss compared to first-order optimizers. However, the generalization properties of second-order methods are still being debated. Theoretical investigations have proved difficult to carry out outside the tractable settings of heavily simplified model classes – thus, the relevance of existing theories to practical deep learning applications remains unclear. Similarly, empirical studies in large-scale models and real datasets are significantly confounded by the necessity to approximate second-order updates in practice. It is often unclear whether the observed generalization behaviour arises specifically from the second-order nature of the parameter updates, or instead reflects the specific structured (e.g. Kronecker) approximations used or any damping-based interpolation towards first-order updates.

Here, we show for the first time that exact Gauss-Newton (GN) updates take on a tractable form in a class of deep reversible architectures that are sufficiently expressive to be meaningfully applied to common benchmark datasets. We exploit this novel setting to study the training and generalization properties of the GN optimizer. We find that exact GN generalizes poorly. In the mini-batch training setting, this manifests as rapidly saturating progress even on the *training* loss, with parameter updates found to overfit each mini-batchatch without producing the features that would support generalization to other mini-batches. We show that our experiments run in the "lazy" regime, in which the neural tangent kernel (NTK) changes very little during the course of training. This behaviour is associated with having no significant changes in neural representations, explaining the lack of generalization.

## 1 Introduction

Efficient optimization of overparameterized neural networks is a major challenge for deep learning. For large models, training remains one of the main computational and time bottlenecks. Much work has therefore been devoted to the development of neural network optimizers that could accelerate training, enabling researchers and engineers to iterate faster and at lower cost in their search for better

---

[*]Equal Contribution. Correspondence to {davide.buffelli,jamie.mcgowan}@mtkresearch.com
[†]Work done while at MediaTek Research.

38th Conference on Neural Information Processing Systems (NeurIPS 2024).

performing models. Second-order optimizers, in particular, have been shown to deliver substantially faster per-iteration progress on the training loss [Martens and Grosse, 2015, Botev et al., 2017, George et al., 2018, Goldfarb et al., 2020, Bae et al., 2022, Petersen et al., 2023, Garcia et al., 2023], and much work has been done to scale them to large models via suitable approximations [Ba et al., 2017, Anil et al., 2021]. However, the generalization properties of second-order optimizers remain poorly understood. Here, we focus on the training and generalization properties of the Gauss-Newton (GN) method, which – in many cases of interest – also encompasses natural gradient descent (NGD) [Martens, 2020].

Theoretical studies of generalization in GN/NGD have been limited to simplified models, such as linear models [Amari et al., 2021] or nonlinear models taken to their NTK limit [Zhang et al., 2019]. When applied to real-world networks and large datasets, GN/NGD has so far required approximations, such as truncated conjugate gradient iterations in matrix-free approaches [Martens et al., 2010], or block-diagonal and Kronecker-factored estimation of the Gauss-Newton / Fisher matrix [Martens and Grosse, 2015, Botev et al., 2017, George et al., 2018, Goldfarb et al., 2020, Bae et al., 2022, Petersen et al., 2023, Garcia et al., 2023]. Those approximations are exact only in the limit of constant NTK [Karakida and Osawa, 2020], in which models cannot learn any features [Yang and Hu, 2021, Aitchison, 2020]. To our knowledge, the only case in which exact and tractable GN updates have been obtained is that of deep linear networks [Bernacchia et al., 2018, Huh, 2020], which – despite exhibiting non-trivial learning dynamics [Saxe et al., 2013] – cannot learn interesting datasets nor yield additional insights into generalization beyond the linear regression setting. Critically, the use of necessary approximations makes it difficult to understand how much of the observed generalization (or lack thereof) can be attributed to the GN method itself, or to the various ways in which it has been simplified.

Here, we derive an exact, computationally tractable expression for Gauss-Newton updates in deep *reversible* networks [Dinh et al., 2015, Mangalam et al., 2022]. In reversible architectures made of stacked, volume-preserving MLP-based coupling layers (which we call RevMLPs), we show that it is possible to analytically derive a specific form of a generalized inverse for the network's Jacobian. This generalized inverse enables fast, exact GN updates in the overparameterized regime. We highlight that, in contrast to the work of Zhang et al. [2019], Cai et al. [2019], Rudner et al. [2019], Karakida and Osawa [2020], we do not assume constant NTK, instead we only require the NTK to remain non-singular during training [Nguyen et al., 2021, Liu et al., 2022, Charles and Papailiopoulos, 2018] as, for example, in the mean-field limit [Arbel et al., 2023]. Equipped with this new model, we study for the first time the generalization behaviour of GN in realistic settings. In the stochastic regime, we find that GN trains *too* well, overfitting single mini-batch at the expense of impaired performance not only on the test set, but also on the training set. To understand this severe lack of generalization, we conduct a careful examination of the model's neural tangent kernel and show that the NTK remains almost unchanged during training, and that the neural representations that arise from after training are not different from those set by the network's initialization. Thus, GN tends to remain in the "lazy" regime [Jacot et al., 2018, Chizat et al., 2019], in which representations remain close to those at initialization, lacking generalization.

In summary:

- We show that GN updates computed with any generalized inverse of the model Jacobian results in the same dynamics of the loss, provided that the NTK does not become singular during training (Theorem 4.3).

- We derive an exact and tractable generalized inverse of the Jacobian in the case of deep reversible neural networks (Proposition 4.4). The corresponding GN updates have the same complexity as gradient descent.

- We study the generalization properties of GN in models up to 147 million parameters on MNIST and CIFAR-10, and we show that neural representations do not change during training, as the model remains in the "lazy" regime.

## 2   Related Work

**Exact vs approximate Gauss-Newton in deep learning.**   Previous work on second-order optimization of deep learning models focused on either Natural Gradient Descent (NGD), or Gauss-Newton

(GN). Since the two are equivalent in many important cases [Martens, 2020], here we do not distinguish them and we refer simply to GN. The most popular methods for computing Gauss-Newton updates assume block-diagonal and Kronecker-factored pre-conditioning matrices [Martens and Grosse, 2015, Botev et al., 2017, George et al., 2018, Goldfarb et al., 2020, Bae et al., 2022, Petersen et al., 2023, Garcia et al., 2023]. Such approximations are known to be exact only in deep linear networks [Bernacchia et al., 2018, Huh, 2020] and in the Neural Tangent Kernel (NTK) limit [Karakida and Osawa, 2020], both of which cannot learn features [Yang and Hu, 2021, Aitchison, 2020]. Recent work focused on exact Gauss-Newton in the feature learning (mean-field) regime [Arbel et al., 2023] but they studied only small models applied to synthetic data. The work of Cai et al. [2019] studies exact Gauss-Newton on real data but only models with one-dimensional outputs. Our work is the first to investigate exact Gauss-Newton in the feature learning regime on real data and sizeable neural networks.

**Reversible neural networks.** Reversible neural networks [Dinh et al., 2015] allow saving memory during training of large models, because they do not require storing activations [Gomez et al., 2017, MacKay et al., 2018], and achieve near state-of-the-art performance [Mangalam et al., 2022]. Reversible neural networks also feature in normalizing likelihood-based generative models, or normalizing flows [Dinh et al., 2016]. In different reversible models, the inverse is either computed analytically with coupling layers [Kingma and Dhariwal, 2018, Chang et al., 2018, Jacobsen et al., 2018] and similar algebraic tricks [Papamakarios et al., 2017, Hoogeboom et al., 2019, Finzi et al., 2019, Xiao and Liu, 2020, Lu and Huang, 2020], is computed numerically [Behrmann et al., 2019, Song et al., 2019, Huang et al., 2020], or is learned [Keller et al., 2021, Teng and Choromanska, 2019]. In this work, we use analytical inversion with coupling layers, because of the efficiency of automatic differentiation through the inverse function. Our work is the first to use reversible neural networks to compute Gauss-Newton updates. A previous work made a connection between reversible models and Gauss-Newton [Meulemans et al., 2020], but they studied Target Propagation, a very different optimizer.

**Generalization of Gauss-Newton in overparameterized models.** The generalization properties of Gauss-Newton are currently debated. While Wilson et al. [2017] shows worst-case scenarios for adaptive methods, Zhang et al. [2019] suggests that GN has similar generalization properties as gradient descent (GD) in the NTK limit. In overparameterized linear models, GN and GD find the same optimum [Amari et al., 2021], however GD transiently achieves better test loss before convergence [Wadia et al., 2021]. The loss dynamics of Gauss-Newton is approximately re-parameterization invariant, and it remains unclear whether a specific parameterizations allows GD to generalize better [Kerekes et al., 2021]. Previous work also suggests a trade-off between training speed and generalization of GN: a good generalization is obtained only when slowing down training, either by damping [Wadia et al., 2021] or by small learning rates [Arbel et al., 2023]. Here we study for the first time generalization for exact GN in sizeable neural networks and real data, and we show that GN achieves poor generalization with respect to gradient descent and similar first order optimizers.

## 3  Background

We provide a brief introduction to Gauss-Newton and Generalized Gauss-Newton. Given input and target data pairs $(x, y) \in \mathbb{R}^{d_x} \times \mathbb{R}^{d_y}$ and parameters $\boldsymbol{\theta} \in \mathbb{R}^p$, the loss is a sum over a batch $\mathcal{B} = \{(x_i, y_i)_{i=1}^n\}$ of $n$ data points

$$\mathcal{L}(\boldsymbol{\theta}) = \sum_{i=1}^n \ell\left(y_i, f(x_i, \boldsymbol{\theta})\right) = \tilde{\mathcal{L}}(\mathbf{f}(\boldsymbol{\theta})) \tag{1}$$

with a twice differentiable and convex function $\ell$ (e.g. square loss or cross entropy) and a parameterized model $f(x_i, \boldsymbol{\theta})$ (e.g. a deep neural network). In the second equality of (1), we concatenate the model outputs $f(x_i, \boldsymbol{\theta}) \in \mathbb{R}^{d_y}$ for all $n$ data points in a single (column) vector $\mathbf{f}(\boldsymbol{\theta}) \in \mathbb{R}^{nd_y}$ with entries $\mathbf{f}_{i+n\cdot(j-1)} = f(x_i, \boldsymbol{\theta})_j$, and define concisely the loss in function space as $\tilde{\mathcal{L}}(\mathbf{f}(\boldsymbol{\theta}))$. The loss $\tilde{\mathcal{L}}(\mathbf{f})$ is a convex and twice differentiable function of the model $\mathbf{f}$, but $\mathcal{L}(\boldsymbol{\theta})$ is usually a non-convex function of the parameters $\boldsymbol{\theta}$, due to the non-linearity of the model $\mathbf{f}(\boldsymbol{\theta})$. Gradient descent optimizes parameters according to:

$$\boldsymbol{\theta}_{t+1} = \boldsymbol{\theta}_t - \alpha \, \nabla_{\boldsymbol{\theta}} \mathcal{L} \tag{2}$$

where $\alpha$ is the learning rate and $\nabla_{\boldsymbol{\theta}}\mathcal{L}$ is the gradient of the loss with respect to the parameters. In the full-batch setting, $\mathcal{B}$ is the full training dataset. In the mini-batch setting (stochastic gradient descent, SGD), a batch of data $\mathcal{B}$ is drawn at random from the dataset at each iteration, without replacement, until all data is covered (one epoch), after which random batches are re-drawn.

**Gauss-Newton.**   We review two alternative but equivalent views on Gauss-Newton: the *Hessian* view and the the *functional* view, which provide different intuitions into the method. The *Hessian* view understands Gauss-Newton as a second-order optimization method, from the point of view of the curvature of the loss. The *functional* view understands Gauss-Newton as model inversion, and is more appropriate in the context of our work.

In the *functional* view, Gauss-Newton corresponds to gradient descent in function space [Zhang et al., 2019, Cai et al., 2019, Bae et al., 2022, Amari, 1998, Martens, 2020]. By assumption, the loss $\tilde{\mathcal{L}}$ is a convex function of the model outputs $\mathbf{f}$, thus it would be convenient to optimize the model outputs directly. Gradient flow in function space is given by

$$\frac{d\mathbf{f}}{dt} = -\nabla_{\mathbf{f}}\tilde{\mathcal{L}}\big|_{\mathbf{f}(t)} \tag{3}$$

However, we need to optimize parameters $\boldsymbol{\theta}$ to have a model that can be applied to new data. If $\mathbf{f}(t) = \mathbf{f}(\boldsymbol{\theta}(t))$, we use the chain rule to find the update in parameter space that corresponds to gradient flow in function space,

$$\frac{\partial\mathbf{f}}{\partial\boldsymbol{\theta}}\frac{d\boldsymbol{\theta}}{dt} = -\nabla_{\mathbf{f}}\tilde{\mathcal{L}}\big|_{\mathbf{f}(\boldsymbol{\theta}(t))} \tag{4}$$

Given the gradient $\nabla_{\mathbf{f}}\tilde{\mathcal{L}}$ and the Jacobian $J = \frac{\partial\mathbf{f}}{\partial\boldsymbol{\theta}}$ defined as $J_{ab} = \frac{\partial\mathbf{f}_a}{\partial\boldsymbol{\theta}_b}$ (of shape $nd_y \times p$), this is a linear system of equations that can be solved for the update $\frac{d\boldsymbol{\theta}}{dt}$, by pseudo-inverting the Jacobian. In discrete time, with learning rate $\alpha$, the update is equal to [Björck, 1996, Ben-Israel, 1965]

$$\boldsymbol{\theta}_{t+1} = \boldsymbol{\theta}_t - \alpha\, J^+\, \nabla_{\mathbf{f}}\tilde{\mathcal{L}} \tag{5}$$

where the superscript $+$ denotes matrix pseudo-inversion. We use this update in our work, in either the full-batch or mini-batch setting. We note that equation (5) implies equation (3), in the continuous time limit, only if the Jacobian has linearly independent rows ($JJ^+ = \mathrm{I}_{nd_y}$), which also guarantees convergence to a global minimum (full-batch). This requires overparameterization $p \geq nd_y$, however, even if the model is underparameterized and does not converge to a global minimum, equation (5) is still equivalent to Gauss-Newton in the *Hessian* view, as shown below.

In the *Hessian* view, Gauss-Newton corresponds to Newton's method with a positive-definite approximation of the Hessian, in the case of square loss [Dennis Jr and Schnabel, 1996, Nocedal and Wright, 1999, Bottou et al., 2018]. The approximation is accurate near a global minimum of the loss, therefore Gauss-Newton inherits the accelerated convergence of Newton's method near global minima [Dennis Jr and Schnabel, 1996]. The Gauss-Newton update, with learning rate $\alpha$, is equal to

$$\boldsymbol{\theta}_{t+1} = \boldsymbol{\theta}_t - \alpha\, \left(J^T J\right)^+ \nabla_{\boldsymbol{\theta}}\mathcal{L} \tag{6}$$

Matrix pseudo-inverse is used instead of inverse when $J^T J$ is singular (damping is also a popular choice, see Nocedal and Wright [1999]). It is straightforward to prove that equations (6) and (5) are identical, by noting that, since $\mathcal{L}(\boldsymbol{\theta}) = \tilde{\mathcal{L}}(\mathbf{f}(\boldsymbol{\theta}))$, then $\nabla_{\boldsymbol{\theta}}\mathcal{L} = J^T\nabla_{\mathbf{f}}\tilde{\mathcal{L}}$ by chain rule, and $\left(J^T J\right)^+ J^T = J^+$ by the properties of matrix pseudo-inverse. The *Gram*-Gauss-Newton update of Cai et al. [2019] is also equivalent to equation (5), it just requires the formula for the Jacobian pseudo-inverse in the case of linearly independent rows.

**Generalized Gauss Newton.**   Following the *Hessian* view, Generalized Gauss-Newton (GGN) was introduced for convex losses that are different from square loss [Ortega and Rheinboldt, 2000, Schraudolph, 2002]. The Hessian is approximated by the positive semi-definite matrix $J^T H J$, where $H = \nabla_{\mathbf{f}}^2\tilde{\mathcal{L}}$. As in the case of square loss, the approximation is accurate near a global minimum. That leads to the following update:

$$\boldsymbol{\theta}_{t+1} = \boldsymbol{\theta}_t - \alpha\, \left(J^T H J\right)^+ \nabla_{\boldsymbol{\theta}}\mathcal{L} \tag{7}$$

Note that GGN reduces to GN for $H = \mathrm{I}_{nd_y}$ (square loss). In the *functional* view, Appendix A shows that Generalized Gauss-Newton corresponds to Newton's method in function space, provided that the Jacobian has linearly independent rows and $\tilde{\mathcal{L}}$ is strongly convex. Furthermore, Appendix B provides some intuition into the convergence of GGN flow.

## 4 Exact and tractable Gauss-Newton

The main hurdle in the GN update of equation (5) is the computation of the Jacobian pseudo-inverse. For a batch size $n$, number of parameters $p$ and output dimension $d$, that requires $\mathcal{O}(ndp \min(nd, p))$ compute and $\mathcal{O}(ndp)$ memory. In this Section, we show that the GN update can be computed efficiently for reversible models. For a dense neural network of $L$ layers and dimension $d$, implying $p = \mathcal{O}(Ld^2)$ parameters, our GN update requires the same memory as gradient descent and $\mathcal{O}(Lnd^2 + Ln^2d)$ compute, compared to $\mathcal{O}(Lnd^2)$ compute of gradient descent.

Our method consists of two steps: first, we replace the Jacobian pseudoinverse with a generalized inverse, and show that it has identical convergence properties (Theorem 4.3). Second, we show that a specific generalized inverse can be computed efficiently in reversible neural networks (Proposition 4.4). We present both results in the case of square loss (GN). Results for other convex loss functions (GGN) can be derived following steps similar to Appendix B.

**Replacing the pseudoinverse with a generalized inverse.** We show that the Jacobian pseudoinverse in equation (5) can be replaced by a generalized inverse that has the same convergence properties. A similar approach was proposed by Bernacchia et al. [2018], Karakida and Osawa [2020], but it was only valid in the case of, respectively, deep linear networks or constant Neural Tangent Kernel (NTK) limit. Here we provide a more general formulation that holds under less restrictive assumptions, e.g. it holds in the mean field regime [Arbel et al., 2023]. We need the following assumption

**Assumption 4.1.** Assume $J(\boldsymbol{\theta})$ has linearly independent rows (is surjective) for all $\boldsymbol{\theta}$ in the domain where GN dynamics takes place.

Note that this implies that the network is overparametrized, i.e. $p \geq nd_y$. While, in practice, this assumption may seem strong, it is only slightly stronger than the following version, employed in Arbel et al. [2023]:

**Assumption 4.2.** $J(\boldsymbol{\theta}_0)$ is surjective at initialization $\boldsymbol{\theta}_0$.

In Arbel et al. [2023], the authors argue that since surjectivity of $J$ is an open condition, it holds for a neighbourhood of $\boldsymbol{\theta}_0$, and moreover continue to prove that the dynamics of GN is well defined up to some exit time $T$ from this neighbourhood. They then continue to give assumptions guaranteeing that this dynamics extends to $\infty$. We directly assume we are in this latter setting.

**Theorem 4.3.** *Under Assumption 4.1 so that there is a right inverse $J^{\dashv}$ satisfying $JJ^{\dashv} = I$, consider the update in parameter space with respect to the flow induced by an arbitrary right inverse $J^{\dashv}$:*

$$\boldsymbol{\theta}_{t+1} = \boldsymbol{\theta}_t - \alpha J^{\dashv} \nabla_{\mathbf{f}} \tilde{\mathcal{L}}. \tag{8}$$

*Then the loss along these trajectories is the same up to $\mathcal{O}(\alpha)$, i.e. for any two choices $J_1^{\dashv}$ and $J_2^{\dashv}$, the corresponding iterates $\boldsymbol{\theta}_t^{(1)}$ and $\boldsymbol{\theta}_t^{(2)}$ satisfy:*

$$\|\nabla_{\mathbf{f}} \tilde{\mathcal{L}}(\mathbf{f}(\boldsymbol{\theta}_t^{(1)})) - \nabla_{\mathbf{f}} \tilde{\mathcal{L}}(\mathbf{f}(\boldsymbol{\theta}_t^{(2)}))\| \leq \mathcal{O}(\alpha). \tag{9}$$

*Moreover, as the Moore-Penrose pseudo-inverse is a right inverse under the assumptions, the result applies to $J^+$, and consequently to the dynamics of (5).*

The proof is in Appendix C. The intuition behind this result becomes clearer once we examine the differential of the loss w.r.t. the function outputs, $\nabla_{\mathbf{f}} \tilde{\mathcal{L}}$. Notice that, as $\tilde{\mathcal{L}}$ is a convex function, it has a unique stationary point, and hence it is natural to interpret $\nabla_{\mathbf{f}} \tilde{\mathcal{L}}(\boldsymbol{\theta})$ as the error at $\boldsymbol{\theta}$, especially close to the global minimum. We will therefore adopt the notation

$$\boldsymbol{\epsilon}(\boldsymbol{\theta}) := \nabla_{\mathbf{f}} \tilde{\mathcal{L}}(\boldsymbol{\theta}) \tag{10}$$

here and throughout the proofs to refer to the deviation from the global minimum at the current parameter value. A key ingredient of the proof of Theorem 4.3 will be to establish that, for trajectories induced by GGN or the update in equation (8), $\boldsymbol{\epsilon}(t) := \boldsymbol{\epsilon}(\boldsymbol{\theta}(t))$ satisfies:

$$\frac{d\boldsymbol{\epsilon}}{dt} = -\boldsymbol{\epsilon}(t) \tag{11}$$

This trivially implies that $\boldsymbol{\epsilon} \to 0$ from any initial condition $\boldsymbol{\epsilon}_0$, so that the evolution of the weights approaches a stationary point for the loss, and hence its global minimum.

The right inverse of the Jacobian, $J^{\dashv}$ is non-unique, and, in general, it is not feasible to compute for large models. However, it turns out that in the case of reversible models, we have an analytic expression for $J^{\dashv}$, which allows computing exact GN at nearly the same cost as SGD.

**Computing GN of a reversible deep network.** Throughout this Section we employ the following notation: for an arbitrary matrix $X$ of shape $(d, n)$ we write the *lowercase boldfont* corresponding symbol, e.g. $\mathbf{x}$ for the row-wise vectorization of the matrix, i.e. $\mathbf{x}_{i+d\cdot(j-1)} = X_{i,j}$.

We consider networks composed of $L$ reversible layers, and we denote by $X_\ell$ (with the associated vectorization $\mathbf{x}_\ell$) and by $W_\ell$ (and $\mathbf{w}_\ell$), respectively, the output and the parameters of layer $\ell$ in matrix and vector forms. The output of the model is the output of the last layer, $\mathbf{f} = \mathbf{x}_L$.

The Jacobian of the full neural network can be expressed as a block matrix consisting of the Jacobians of different layers. Letting $\boldsymbol{\theta} = (\mathbf{w}_1, \mathbf{w}_2, \ldots, \mathbf{w}_L)$ the concatenated vector with parameters of all layers

$$J = \frac{\partial \mathbf{x}_L}{\partial(\mathbf{w}_1, \ldots, \mathbf{w}_L)} = (J_1, \ldots, J_L) \tag{12}$$

with $J_\ell = \frac{\partial \mathbf{x}_L}{\partial \mathbf{w}_\ell}$. Since the only way $\mathbf{w}_\ell$ affects $\mathbf{x}_L$ is through the way it affects $\mathbf{x}_\ell$, by the chain rule, the layer-wise Jacobian can be written as

$$J_\ell = \frac{\partial \mathbf{x}_L}{\partial \mathbf{x}_\ell}\frac{\partial \mathbf{x}_\ell}{\partial \mathbf{w}_\ell} \tag{13}$$

First, we note that a right inverse of the full Jacobian in equation (12) can be computed by finding right inverses of the individual, layer-wise Jacobians of equation (13). Then we show that, given that the neural network is reversible, a right inverse of equation (13) can be computed easily. In particular, the product of the inverse of the first factor $\partial \mathbf{x}_L / \partial \mathbf{x}_\ell$ with any vector can be computed exactly with a single forward differentiation pass on the neural network's inverse. The inverse of the second factor $\partial \mathbf{x}_\ell / \partial \mathbf{w}_\ell$ can be also computed at low complexity when individual layers are linear in the parameters, even if nonlinear in the input. These observations hold for any reversible neural network, but here we use dense coupling layers as a specific realization (see Section 2), which we call RevMLP. The activation $\mathbf{x}_\ell \in \mathbb{R}^{dn}$ for layer $\ell$ is written in matrix form $X_\ell \in \mathbb{R}^{d \times n}$ and is split along the first dimension into two components $X_\ell = (X_\ell^{(1)}, X_\ell^{(2)})$, where $X_0$ is the input. Here $d$ is an even integer and both $X_\ell^{(1)}$ and $X_\ell^{(2)}$ have shape $\left(\frac{d}{2} \times n\right)$. The equations for a single coupling layer are

$$X_\ell^{(1)} = X_{\ell-1}^{(1)} + W_\ell^{(1)}\sigma(V_{\ell-1}^{(2)}X_{\ell-1}^{(2)}) \tag{14}$$

$$X_\ell^{(2)} = X_{\ell-1}^{(2)} + W_\ell^{(2)}\sigma(V_\ell^{(1)}X_\ell^{(1)}) \tag{15}$$

where $W_\ell = (W_\ell^{(1)}, W_\ell^{(2)})$ are trainable parameters, while $V_\ell = (V_\ell^{(1)}, V_\ell^{(2)})$ are *non*-trainable parameters (also known as *inverted bottleneck*, see Bachmann et al. [2024]), and $\sigma(\cdot)$ is any differentiable non-linear function. In the rest of this paper, we use the term *layer* and *block* interchangeably to refer to a full coupling layer (i.e., where the output is the concatenation of $X_\ell^{(1)}$ and $X_\ell^{(2)}$ as defined in Equation (14) and Equation (15)). Whereas we explicitly refer to "half"-coupled layers to mean Equation (14) or Equation (15). We also define the reshaping operator: for $\mathbf{x}$, a vector of $dn$ components we write $\mathcal{R}^{(d,n)}\{\mathbf{x}\}$ for the matrix $A$ of size $(d \times n)$ satisfying: $A_{i,j} = \mathbf{x}_{i+d(j-1)}$

**Proposition 4.4.** *Assuming $\sigma(V_{\ell-1}^{(2)}X_{\ell-1}^{(2)})$, $\sigma(V_\ell^{(1)}X_\ell^{(1)})$ have linearly independent columns, the GN update for the weights of each layer is given by*

$$W_\ell^{(1)}(t+1) = W_\ell^{(1)}(t) - \frac{\alpha}{L} \underbrace{\mathcal{R}^{(\frac{d}{2},n)}\left\{\frac{\partial \mathbf{x}_\ell^{(1)}}{\partial \mathbf{x}_L}\boldsymbol{\epsilon}\right\}\sigma\left(V_{\ell-1}^{(2)}X_{\ell-1}^{(2)}\right)^+}_{\Delta_\ell^{(1)}} \tag{16}$$

$$W_\ell^{(2)}(t+1) = W_\ell^{(2)}(t) - \frac{\alpha}{L}\mathcal{R}^{(\frac{d}{2},n)}\left\{\frac{\partial \mathbf{x}_\ell^{(2)}}{\partial \mathbf{x}_L}\boldsymbol{\epsilon} - \left(\frac{\partial \mathbf{x}_\ell^{(2)}}{\partial \mathbf{w}_\ell^{(1)}}\right)\mathcal{R}^{(\frac{d}{2},d')^{-1}}\left\{\Delta_\ell^{(1)}\right\}\right\}\sigma\left(V_\ell^{(1)}X_\ell^{(1)}\right)^+ \tag{17}$$

The proof is in Appendix D.

**Computational and Memory Complexity**

The terms in the braces in equations (16), (17), are Jacobian-vector products and can be easily computed using automatic mode differentiation at the cost of one (reverse) inference pass, i.e., $\mathcal{O}(nd^2)$.

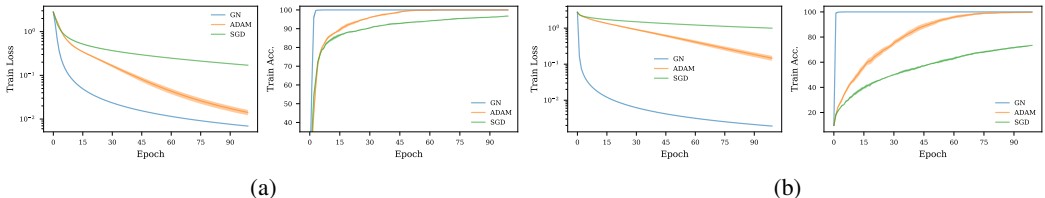

(a)                                                                          (b)

Figure 1: Training loss and accuracy on (a) MNIST and (b) CIFAR-10 in a full-batch scenario where each dataset is trimmed to a fixed subset of $n = 1024$ images. GN converges much faster than Adam and SGD.

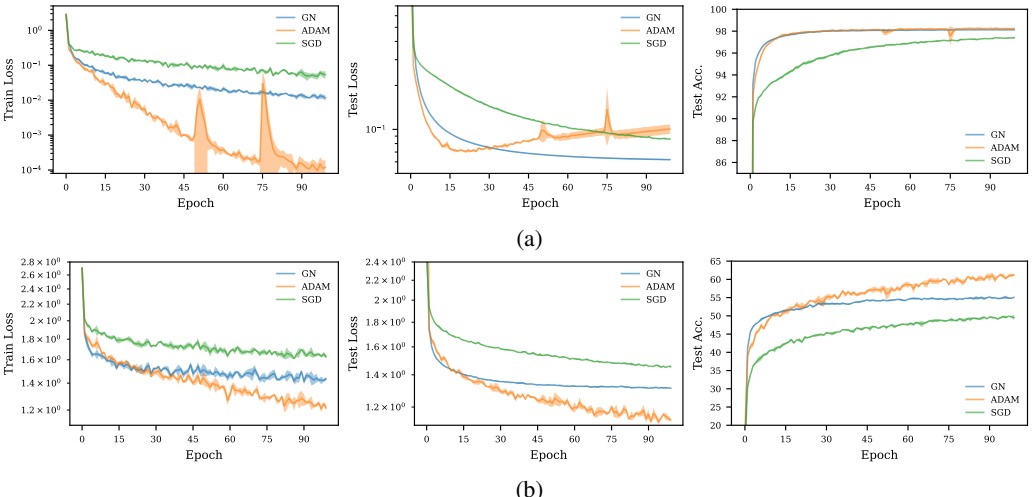

(a)

(b)

Figure 2: Training loss, test loss, and test accuracy on (a) MNIST and (b) CIFAR-10 in a mini-batch scenario. GN does not exhibit the same properties observed in the full-batch setting. In fact, Adam reaches lower training and test loss.

The last factor requires pseudo-inverting a matrix of size $n \times d/2$, which requires $\mathcal{O}(nd \min(n, d))$. Since these operations are required in each layer, the overall cost of the update for the full network is $\mathcal{O}(Lnd^2 + Ln^2d)$, compared to $\mathcal{O}(Lnd^2)$ of SGD, while the memory complexity is the same.

## 5   Experiments

For our experiments we train RevMLPs (equations (14) and (15)) with 2 (6) blocks for MNIST [LeCun et al., 2010] (CIFAR-10; Krizhevsky, 2009), ReLU non-linearities at all half-coupled layers, and an inverted bottleneck of size 8000 resulting in models with 12M (MNIST) and 147M (CIFAR-10) parameters. We train these models to classify images flattened to 1D vectors, using a cross-entropy objective. Note that the chosen size of the inverted bottleneck ensures that the assumptions of Proposition 4.4 hold.

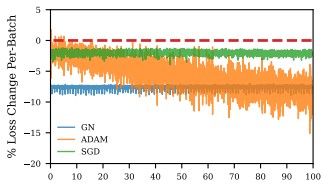

Figure 3: Percentage change in training loss after each update. GN decreases the loss for the current mini-batch more than SGD and Adam early in training.

At each training iteration, we compute the pseudoinverses in equations (16), (17) using an SVD. For numerical stability we truncate the SVD to a 1% tolerance relative to the largest singular value and an absolute tolerance of $10^{-5}$, whichever gives the smallest rank – our main findings are qualitatively robust to these tolerance levels. Full hyperparameters and additional details are reported in Appendix L, and code is provided with the submission. We report results averaged over 3 random seeds. All experiments are performed on a single NVIDIA RTXA6000 GPU.

## 5.1 Full-Batch Setting

We first examine the full-batch setting in which the dataset is a random subset of size 1024 of MNIST or CIFAR-10. We tuned the learning rate for each optimizer by selecting the largest one that did not cause the loss to diverge. Figure 1 shows that GN is significantly faster than Adam and SGD in both datasets, in line with theoretical predictions (Equation 11).

## 5.2 Mini-Batch Setting

Next, we consider the full MNIST and CIFAR-10 datasets in the mini-batch setting. We follow standard train and test splits for the datasets, with a mini-batch size of $n = 1024$. The learning rate for all methods is tuned towards the largest progress after 1 epoch that does not exhibit any training instabilities. In both datasets, GN makes good initial progress on the training and test losses, but then struggles to sustain the continued progress which Adam exhibits (Fig. 2). This surprising early saturation of the GN training and test losses is most pronounced for the CIFAR dataset, where even SGD eventually overtakes GN (see Fig. 6 in Appendix E for a longer training run). In the rest of this Section, we use the CIFAR-10 setup to study the possible origins of such weak generalization.

**Overfitting each mini-batch.** Based on the full-batch results of Figure 1 in which GN was seen to converge very fast, we postulate that the poor generalization behaviour observed in the mini-batch case may be caused by overfitting to each mini-batch. To test this hypothesis, at each iteration, we compute the loss on a single mini-batch before and after applying the update computed on that same mini-batch. The resulting percentage change in mini-batch loss is shown in Figure 3. Compared to SGD and Adam, GN leads to a much stronger immediate decrease in loss after each update, especially early in training. Whilst this difference gradually weakens during the course of training, it subsists for 80 epochs, i.e. until well after GN's overall training and test losses have saturated (c.f. Fig.2). These results suggest that GN might require some form of regularization to prevent aggressive incorporation of each mini-batch into the model's parameters. However, we find that neither smaller learning rates (Appendix I), nor weight decay (Appendix J), nor any of the usual techniques for regularizing the pseudoinverse in Equation (16) (Appendix K) appear to help in this respect (Figures 12, 13 and 14).

**Evolution of the Neural Tangent Kernel.** We further hypothesize that GN's poor generalization may be due to a lack of feature learning. In a similar fashion to Fort et al. [2020], we study the evolution of the neural tangent kernel (NTK) when training with GN compared to SGD and Adam. A changing NTK would suggest that the model learns features different from those present at initialization. Figure 4a and Figure 4b show the rate of change of the NTK between epochs, and the evolution of the NTK similarity with initialization, respectively.

Overall, the NTK changes very little for SGD and GN, suggesting that SGD and GN operate close to the "lazy" training regime [Jacot et al., 2018, Chizat et al., 2019]. On the other side, Adam causes the NTK to change significantly, i.e. Adam does learn features different from the initial ones.

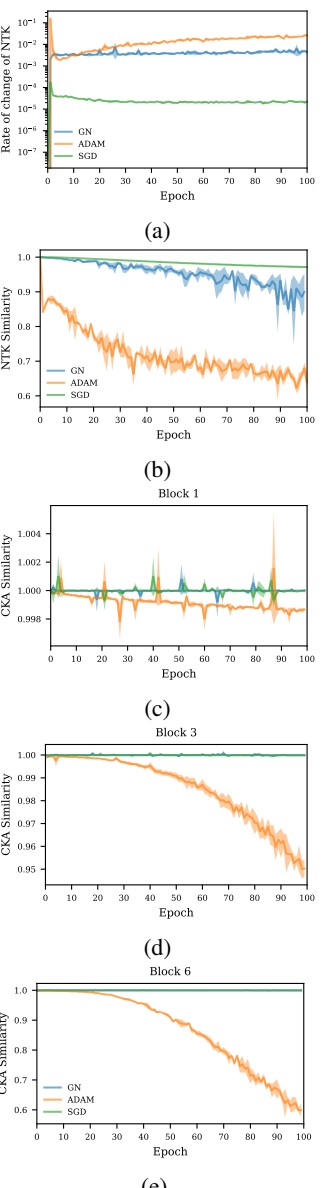

Figure 4: NTK and CKA similarity evolution across training for GN, Adam and SGD. Top two panels include (a) the rate of change of the NTK and (b) the NTK similarity during training with respect to initialization. Bottom three panels, along the same axis, include the CKA similarities for the (c) first, (d) middle and (e) last block with respect to their initial values.

**Feature Learning with Gauss-Newton.** Even if features (i.e., the NTK) change during GN training, it remains unclear whether they are associated with changes in neural representations. We examine the change in neural representations during training by computing the Centered Kernel Alignment (CKA; Kornblith et al., 2019 ) measure of similarity between the representations at initialization and those learned at each epoch, across all layers of the model.

Figures 4c, 4d and 4e illustrate the evolution of CKA similarities for the last, middle and first block (i.e., a "full" coupling layer as described by equations 14, 15) of a 12 layer RevMLP trained on CIFAR-10. Plots for all blocks are provided in Appendix H along with pairwise similarities across optimizers (Figures 10 and 11). Similar to the NTK, neural representations do not change during training with GN. SGD behaves similarly, with little change in CKA. Adam, on the contrary, has changes in neural representations that coincide with changes in NTK. Appendix G shows that the lack of changes in neural representations for GN cannot be explained by a smaller change in parameters, in fact both GN and Adam show changes in weight space, while weights of SGD change little (Figure 9).

Contrary to the findings of Arbel et al. [2023], it is evident that the CKA similarities in Figures 4c, 4d and 4e remain higher for longer in earlier blocks for GN, implying that GN is slower than Adam at adapting its deeper internal representations. Furthermore, in Appendix F and I, we address two suggestions from Arbel et al. [2023] and find that the generalization improvements when using smaller learning rates and/or different initializations (close-to-optimal in Figure 7 and low variance in Figure 8) do not carry over to deeper networks. In particular, Figure 7 shows that continuing training with GN after initially training with Adam exhibits the same phenomena as training with GN throughout – albeit at a slightly lower loss than can be achieved using only GN.

## 5.3 Experiments without Inverted Bottleneck

The previous experiments used inverted bottlenecks to ensure "linear independence", i.e., to ensure that the model is adequately overparametrized such that the proposed efficient generalized inverse is valid. In other words, inverted bottlenecks ensure that the scalable GN weight updates (equations 16) do implement gradient flow in function space (equation 3), such that our results are not potentially confounded by broken theoretical assumptions. Nevertheless, the proposed GN update can still be applied in the absence of inverted bottlenecks. In Figure 5 we report results on the CIFAR10 dataset, following the same experimental procedure of the previous experiments, but removing all inverted bottlenecks. In the full-batch setting, GN is still performing much better than Adam and SGD. In the mini-batch setting we observe a very similar trend to what is observed in the previous experiments: GN leads to an early saturation of the loss, which instead does not appear in Adam and SGD.

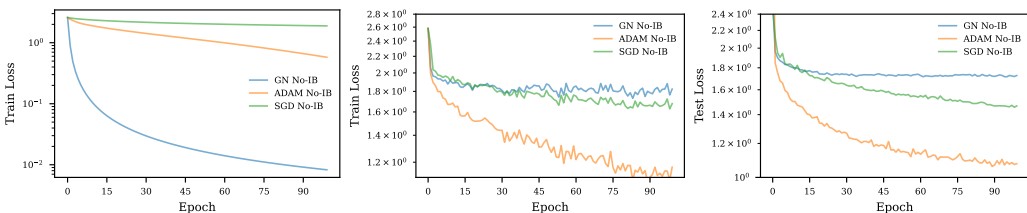

Figure 5: Experiments on CIFAR using a model without Inverted Bottleneck (Full-batch on the left, mini-batch on center and right). While the theoretical guarantees do not hold in this setting, the results follow the same trend observed in Figure 2.

## 5.4 Regression Experiments

We further performed some experiments on regression tasks from the UCI regression dataset[†]. In more detail, we used the Superconductivity Hamidieh [2018] and Wine Aeberhard and Forina [1992] datasets, and followed the same experimental procedure used for the classification datasets (i.e., we use the same RevMLP architecture with an inverted bottleneck for all optimizers and select the highest learning rate that does not cause the loss to diverge). Results are shown in Appendix M and follow the same trend observed in the classification experiments: in the full-batch case GN is

---

[†]`https://github.com/treforevans/uci_datasets`

significantly faster than SGD and Adam, while in the mini-batch case there is an apparent stagnation of the test and train losses under GN.

## 6   Summary and limitations

In this paper we have introduced a new, tractable way of computing exact Gauss-Newton updates in models with millions of parameters. We have used this theory to study the generalization behaviour of GN in realistic task settings. We found that, although GN yields fast convergence in the full batch regime as predicted, it does not perform as well in the stochastic setting where it tends to overfit each mini-batch. We observed that the NTK does not change when training with GN, suggesting that it operates in the "lazy" regime. In line with the above, using the CKA metric, we performed an analysis of the neural representations at the start and end of training showing that they remain very close to each other. This can explain the observed lack of generalization.

Our investigations have relied on a specific formulation of GN based on a tractable generalized inverse of the Jacobian in reversible networks. While we proved that this inverse leads to the same training loss dynamics, in the limit of small learning rate, as the standard GN formulation is based on the Moore-Penrose pseudoinverse (MPP), one cannot exclude the possibility that those two update rules have different learning and generalization properties for finite learning rates. Indeed, the functional view of GN (Section 3) makes it clear that the standard MPP-based GN update corresponds to the minimum-norm weight update that guarantees (infinitesimal) steepest descent in function space. Whilst also achieving steepest descent, our generalized inverse does not have the same least-squares interpretation – although it could imply another form of regularization which future work could uncover. In any case, these differences are difficult to assess precisely because the full Jacobian of the network (let alone its MPP) simply cannot be computed for large models.

Previous applications of approximate GN to deep models found that damping, or truncating, the pseudoinverse of the GN matrix (or, equivalently, of the Jacobian) is key not only for good generalization but also for successful training [Martens et al., 2010, Wadia et al., 2021]. Our generalized inverse is based on a layer-wise factorization of the Jacobian, teasing apart (i) the sensitivity of the network's output to small changes in layer activations and (ii) the sensitivity of those activations to small changes in parameters. The use of exactly reversible networks allows us to invert the former very efficiently, but does not easily accommodate damping or truncation, making it difficult to study their effect on generalization in large scale settings. We speculate that a variation on coupling layer-based reversible networks could be developed that allows for damping or truncation, potentially improving the generalization behaviour of GN. If this can be done, our framework would then enable very efficient training of large models, effectively achieving the training acceleration of second-order methods at the cost of first-order optimizers, all in a memory efficient architecture.

## Acknowledgments and Disclosure of Funding

The authors would like to thank Emmeran Johnson for proving that our layer-wise right inverse of the Jacobian (Equation (38)) is actually the Moore-Penrose pseudoinverse.

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

# Appendix

## A   Functional view of Generalized Gauss-Newton

Assuming that $\tilde{\mathcal{L}}$ is strongly convex, Newton's flow in function space is equal to

$$\frac{d\mathbf{f}}{dt} = -H^{-1}\,\nabla_{\mathbf{f}}\tilde{\mathcal{L}} \tag{18}$$

with $H = \nabla_{\mathbf{f}}^2\tilde{\mathcal{L}}$. Following steps similar to Section 3, we use $\frac{d\mathbf{f}}{dt} = J\frac{d\boldsymbol{\theta}}{dt}$ and pseudo-invert the Jacobian. In discrete time, with learning rate $\alpha$, the functional view of Generalized Gauss Newton is given by

$$\boldsymbol{\theta}_{t+1} = \boldsymbol{\theta}_t - \alpha\,J^+H^{-1}\nabla_{\mathbf{f}}\tilde{\mathcal{L}} \tag{19}$$

It is straightforward to show that equations (19) and (7) are identical when the Jacobian has linearly independent rows. Under these assumptions, $\left(J^THJ\right)^+ = J^+H^{-1}J^{T^+}$ and $J^{T^+}J^T = I$. Furthermore, as in Section 3, $\nabla_{\boldsymbol{\theta}}\mathcal{L} = J^T\nabla_{\mathbf{f}}\tilde{\mathcal{L}}$.

## B   Convergence of GGN flow

In this Section, we give an informal derivation of the continuous-time dynamics of GGN. See Ortega and Rheinboldt [2000], Bottou et al. [2018] for convergence of GGN in discrete time. We consider the optimization of the error under GGN flow in continuous time. Using the definition of the error $\boldsymbol{\epsilon} = \nabla_{\mathbf{f}}\tilde{\mathcal{L}}$ (equation (10)), the optimization flow of the error can be derived using the chain rule

$$\frac{d\boldsymbol{\epsilon}}{dt} = HJ\frac{d\boldsymbol{\theta}}{dt} \tag{20}$$

The definition of GGN flow is

$$\frac{d\boldsymbol{\theta}}{dt} = -\alpha\,(J^THJ)^+\nabla_{\boldsymbol{\theta}}\mathcal{L} = -\alpha\,(J^THJ)^+J^T\boldsymbol{\epsilon} \tag{21}$$

Therefore, optimization of the error under GGN flow is equal to

$$\frac{d\boldsymbol{\epsilon}}{dt} = -\alpha\,HJ(J^THJ)^+J^T\boldsymbol{\epsilon} = -\alpha\,A\boldsymbol{\epsilon} \tag{22}$$

where we defined the matrix $A = HJ(J^THJ)^+J^T$. Using the properties of the matrix pseudoinverse, we note that $A$ is a projection operator, namely $A^n = A$ for any integer power $n$. Therefore, eigenvalues of $A$ are either zero or one, implying that the error decreases exponentially in the range of $A$, while it remains constant in the null space of $A$. In general, the range and null space of $A$ change during training. We note that the projection is orthogonal with respect to the inner product $\boldsymbol{\epsilon}^T H\boldsymbol{\epsilon}$. Using the same steps as in Appendix A, if $J$ has linearly independent rows and $\tilde{\mathcal{L}}$ is strongly convex, then it is straightforward to show that $A = \mathrm{I}_{nd_y}$.

## C   Proof of Theorem 4.3

During the proof we will compare the dynamics of the flow curves of two vector fields, namely $-J^+(\boldsymbol{\theta})\boldsymbol{\epsilon}(\boldsymbol{\theta})$ and the corresponding $-J^{\dashv}(\boldsymbol{\theta})\boldsymbol{\epsilon}(\boldsymbol{\theta})$. The dependence on $\boldsymbol{\theta}$ is assumed throughout and we will write $(-J^+\boldsymbol{\epsilon})\big|_{\boldsymbol{\theta}}$ or just $-J^+\boldsymbol{\epsilon}$. The flowlines of these vector fields are given by:

$$\frac{d\boldsymbol{\theta}}{dt} = (-J^+\boldsymbol{\epsilon})\big|_{\boldsymbol{\theta}(t)} \tag{23}$$

and

$$\frac{d\boldsymbol{\theta}}{dt} = (-J^{\dashv}\boldsymbol{\epsilon})\big|_{\boldsymbol{\theta}(t)} \tag{24}$$

respectively. We will further write plain $\boldsymbol{\theta}(t)$ for the solutions of equation (23) and $\tilde{\boldsymbol{\theta}}(t)$ for the solutions of equation (24). Moreover, we'll write $\tilde{\mathbf{f}}$ to mean $\mathbf{f}(\tilde{\boldsymbol{\theta}}(t))$ for arbitrary (possibly tensor valued) functions $\mathbf{f}$, to distinguish from $\mathbf{f}(\boldsymbol{\theta}(t))$.

First notice that the right inverses $J^{\dashv}$ are not canonical and hence equation (24) represents a family of equations and associated flowlines. However, the dynamics of the associated error $\tilde{\epsilon}$ is the same regardless of the choice, and moreover, the same as that of $\epsilon$ itself. Note that $\frac{d}{dt}\epsilon(\boldsymbol{\theta}(t)) = J\big|_{\boldsymbol{\theta}(t)} \cdot \frac{d\boldsymbol{\theta}}{dt}$, which gives:

$$\frac{d\epsilon}{dt} = -JJ^+\epsilon \tag{25}$$

If $J$ has linearly independent rows we have $JJ^+ = \mathrm{I}$, therefore

$$\frac{d\epsilon}{dt} = -\epsilon \tag{26}$$

If we consider the flow of $\tilde{\epsilon} := \epsilon(\tilde{\boldsymbol{\theta}}(t))$, replacing $J^+$ with a right inverse $J^{\dashv}$ satisfies $JJ^{\dashv} = \mathrm{I}$, and again we obtain

$$\frac{d\tilde{\epsilon}}{dt} = -\tilde{\epsilon} \tag{27}$$

Integrating these equations from the same initial condition $\tilde{\boldsymbol{\theta}}(t_0) = \boldsymbol{\theta}(t_0)$ indeed gives the same error flowlines.

So despite the different dynamics of the $\boldsymbol{\theta}$ and $\tilde{\boldsymbol{\theta}}$, errors propagate identically. We can use this to derive a bound between the errors incurred by the $k$-th forward Euler iterates of equations (23) and (24), which define the gradient descent equations. Denote by $\boldsymbol{\theta}_i$ the $i$-th Euler iterate of $\boldsymbol{\theta}(t)$ and by $\tilde{\boldsymbol{\theta}}_i$, the corresponding iterate of $\tilde{\boldsymbol{\theta}}(t)$. Then, we have:

$$\|\boldsymbol{\theta}_i - \boldsymbol{\theta}(t_i)\| \leq \frac{\alpha}{K}\left(e^{L(t_i-t_0)} - 1\right) \tag{28}$$

$$\|\tilde{\boldsymbol{\theta}}_i - \tilde{\boldsymbol{\theta}}(t_i)\| \leq \frac{\alpha}{\tilde{K}}\left(e^{\tilde{L}(t_i-t_0)} - 1\right) \tag{29}$$

where $\alpha$ is the step-size, $i$, the number of steps can be computed as $i = \lfloor \frac{t_i-t_0}{\alpha} \rfloor$, $L$ and $\tilde{L}$ are the Lipschitz constants of $(-J^+\epsilon)$ and $(-J^{\dashv}\epsilon)$ respectively, and $K$ and $\tilde{K}$ are constants which depend on the maximum norm of $\frac{d^2}{dt^2}\boldsymbol{\theta}(t)$ and $\frac{d^2}{dt^2}\boldsymbol{\theta}(t)$ across our domain, see e.g. Iserles. Since $\epsilon$ is at least $\mathcal{C}^2$ and the dynamics of $\boldsymbol{\theta}$ and $\tilde{\boldsymbol{\theta}}$ take place over a bounded domain, $\epsilon$ is Lipschitz with constant $L_\epsilon$. Then we have:

$$\|\epsilon(\boldsymbol{\theta}_i) - \epsilon(\tilde{\boldsymbol{\theta}}_i)\| \leq \|\epsilon(\boldsymbol{\theta}_i) - \epsilon(\boldsymbol{\theta}(t_i))\| + \|\epsilon(\boldsymbol{\theta}(t_i)) - \epsilon(\tilde{\boldsymbol{\theta}}(t_i))\| + \|\epsilon(\tilde{\boldsymbol{\theta}}(t_i)) - \epsilon(\tilde{\boldsymbol{\theta}}_i)\| \tag{30}$$

$$\leq L_\epsilon\|\boldsymbol{\theta}_i - \boldsymbol{\theta}(t_i)\| + 0 + L_\epsilon\|\tilde{\boldsymbol{\theta}}_i - \tilde{\boldsymbol{\theta}}(t_i)\| \tag{31}$$

$$\leq L_\epsilon\frac{\alpha}{\bar{K}}\left(e^{\bar{L}(t_i-t_0)} - 1\right) \tag{32}$$

where $\bar{K} = \min(K, \tilde{K})$ and $\bar{L} = \max(L, \tilde{L})$. This shows that the dynamics of the gradient descent iterates coincides up to first order in $\alpha$.

## D  Proof of Proposition 4.4

*Proof.* We rewrite the layer-wise Jacobian as

$$J_\ell = \frac{\partial \mathbf{x}_L}{\partial \mathbf{x}_\ell}\frac{\partial \mathbf{x}_\ell}{\partial \mathbf{w}_\ell} = \frac{\partial \mathbf{x}_L}{\partial(\mathbf{x}_\ell^{(1)}, \mathbf{x}_\ell^{(2)})}\frac{\partial(\mathbf{x}_\ell^{(1)}, \mathbf{x}_\ell^{(2)})}{\partial(\mathbf{w}_\ell^{(1)}, \mathbf{w}_\ell^{(2)})} \tag{33}$$

$$= \left(\frac{\partial \mathbf{x}_L}{\partial \mathbf{x}_\ell^{(1)}}, \frac{\partial \mathbf{x}_L}{\partial \mathbf{x}_\ell^{(2)}}\right)\begin{pmatrix} \sigma(V_{\ell-1}^{(2)}X_{\ell-1}^{(2)})^T \otimes \mathrm{I}_{d/2} & 0 \\ \frac{\partial \mathbf{x}_\ell^{(2)}}{\partial \mathbf{w}_\ell^{(1)}} & \sigma(V_\ell^{(1)}X_\ell^{(1)})^T \otimes \mathrm{I}_{d/2} \end{pmatrix} \tag{34}$$

$$= \left(\frac{\partial \mathbf{x}_L}{\partial \mathbf{x}_\ell^{(1)}}, \frac{\partial \mathbf{x}_L}{\partial \mathbf{x}_\ell^{(2)}}\right)\begin{pmatrix} \sigma_{\ell-1}^{(2)}{}^T \otimes \mathrm{I}_{d/2} & 0 \\ \frac{\partial \mathbf{x}_\ell^{(2)}}{\partial \mathbf{w}_\ell^{(1)}} & \sigma_\ell^{(1)}{}^T \otimes \mathrm{I}_{d/2} \end{pmatrix} \tag{35}$$

where $\sigma_{\ell-1}^{(2)} = \sigma(V_{\ell-1}^{(2)} X_{\ell-1}^{(2)})$ and $\sigma_{\ell}^{(1)} = \sigma(V_{\ell}^{(1)} X_{\ell}^{(1)})$ for brevity.

Given a lower-triangular block matrix

$$\begin{pmatrix} A & 0 \\ B & C \end{pmatrix} \tag{36}$$

it is possible to define a right inverse as

$$\begin{pmatrix} A^+ & 0 \\ -C^+ B A^+ & C^+ \end{pmatrix} \tag{37}$$

Using the above, we prove that a right inverse $J_\ell^{\dashv}$ of Equation (34) is equal to

$$J_\ell^{\dashv} = \begin{pmatrix} \left[ \sigma_{\ell-1}^{(2)}{}^{T+} \otimes \mathbf{I}_{d/2} \right] & 0 \\ -\left[ \sigma_{\ell}^{(1)}{}^{T+} \otimes \mathbf{I}_{d/2} \right] \frac{\partial \mathbf{x}_{\ell}^{(2)}}{\partial \mathbf{w}_{\ell}^{(1)}} \left[ \sigma_{\ell-1}^{(2)}{}^{T+} \otimes \mathbf{I}_{d/2} \right] & \left[ \sigma_{\ell}^{(1)}{}^{T+} \otimes \mathbf{I}_{d/2} \right] \end{pmatrix} \begin{pmatrix} \frac{\partial \mathbf{x}_{\ell}^{(1)}}{\partial \mathbf{x}_L} \\ \frac{\partial \mathbf{x}_{\ell}^{(2)}}{\partial \mathbf{x}_L} \end{pmatrix} \tag{38}$$

It is possible to prove that $J_\ell^{\dashv}$ defined above in Equation 38 actually corresponds to the Moore-Penrose pseudoinverse of $J_\ell$ (see Appendix N). From Equation (34) and Equation (38), we have that

$$J_\ell J_\ell^{\dashv} = \left( \frac{\partial \mathbf{x}_L}{\partial \mathbf{x}_{\ell}^{(1)}}, \frac{\partial \mathbf{x}_L}{\partial \mathbf{x}_{\ell}^{(2)}} \right) \begin{pmatrix} \left[ \sigma_{\ell-1}^{(2)} \sigma_{\ell-1}^{(2)}{}^+ \right]^T \otimes \mathbf{I}_{d/2} & 0 \\ \Lambda & \left[ \sigma_{\ell}^{(1)} \sigma_{\ell}^{(1)}{}^+ \right]^T \otimes \mathbf{I}_{d/2} \end{pmatrix} \begin{pmatrix} \frac{\partial \mathbf{x}_{\ell}^{(1)}}{\partial \mathbf{x}_L} \\ \frac{\partial \mathbf{x}_{\ell}^{(2)}}{\partial \mathbf{x}_L} \end{pmatrix} \tag{39}$$

$$= \left( \frac{\partial \mathbf{x}_L}{\partial \mathbf{x}_{\ell}^{(1)}}, \frac{\partial \mathbf{x}_L}{\partial \mathbf{x}_{\ell}^{(2)}} \right) \begin{pmatrix} I & 0 \\ 0 & I \end{pmatrix} \begin{pmatrix} \frac{\partial \mathbf{x}_{\ell}^{(1)}}{\partial \mathbf{x}_L} \\ \frac{\partial \mathbf{x}_{\ell}^{(2)}}{\partial \mathbf{x}_L} \end{pmatrix} \tag{40}$$

$$= \frac{\partial \mathbf{x}_L}{\partial \mathbf{x}_{\ell}^{(1)}} \frac{\partial \mathbf{x}_{\ell}^{(1)}}{\partial \mathbf{x}_L} + \frac{\partial \mathbf{x}_L}{\partial \mathbf{x}_{\ell}^{(2)}} \frac{\partial \mathbf{x}_{\ell}^{(2)}}{\partial \mathbf{x}_L} = \frac{\partial \mathbf{x}_L}{\partial (\mathbf{x}_{\ell}^{(1)}, \mathbf{x}_{\ell}^{(2)})} \frac{\partial (\mathbf{x}_{\ell}^{(1)}, \mathbf{x}_{\ell}^{(2)})}{\partial \mathbf{x}_L} = \frac{\partial \mathbf{x}_L}{\partial \mathbf{x}_{\ell}} \frac{\partial \mathbf{x}_{\ell}}{\partial \mathbf{x}_L} = \mathbf{I}_{dn} \tag{41}$$

where Equation (40) follows from the assumption that $\sigma_{\ell-1}^{(2)}$ and $\sigma_{\ell}^{(1)}$ have linearly independent columns and,

$$\Lambda = \frac{\partial \mathbf{x}_{\ell}^{(2)}}{\partial \mathbf{w}_{\ell}^{(1)}} \left[ \sigma_{\ell-1}^{(2)}{}^{T+} \otimes \mathbf{I}_{d/2} \right] - \left( \left[ \sigma_{\ell}^{(1)} \sigma_{\ell}^{(1)}{}^+ \right]^T \otimes \mathbf{I}_{d/2} \right) \frac{\partial \mathbf{x}_{\ell}^{(2)}}{\partial \mathbf{w}_{\ell}^{(1)}} \left[ \sigma_{\ell-1}^{(2)}{}^{T+} \otimes \mathbf{I}_{d/2} \right] = 0. \tag{42}$$

Additionally, the last equality in Equation (41) holds since $\mathbf{x}_L$ is a bijective function of $\mathbf{x}_\ell$, due to the reversibility of the RevMLP. Finally, following from Equation (12), we note that,

$$J^{\dashv} = \frac{1}{L} \begin{pmatrix} J_1^{\dashv} \\ \vdots \\ J_L^{\dashv} \end{pmatrix} \tag{43}$$

which when substituted into Equation (8), along with Equation (38), results in the GN update for the weights of each layer,

$$W_{\ell}^{(1)}(t+1) = W_{\ell}^{(1)}(t) - \frac{\alpha}{L} \underbrace{\mathcal{R}^{(\frac{d}{2}, n)} \left\{ \frac{\partial \mathbf{x}_{\ell}^{(1)}}{\partial \mathbf{x}_L} \boldsymbol{\epsilon} \right\} \sigma \left( V_{\ell-1}^{(2)} X_{\ell-1}^{(2)} \right)^+}_{\Delta_{\ell}^{(1)}} \tag{44}$$

$$W_{\ell}^{(2)}(t+1) = W_{\ell}^{(2)}(t) - \frac{\alpha}{L} \mathcal{R}^{(\frac{d}{2}, n)} \left\{ \frac{\partial \mathbf{x}_{\ell}^{(2)}}{\partial \mathbf{x}_L} \boldsymbol{\epsilon} - \left( \frac{\partial \mathbf{x}_{\ell}^{(2)}}{\partial \mathbf{w}_{\ell}^{(1)}} \right) \mathcal{R}^{(\frac{d}{2}, d')^{-1}} \left\{ \Delta_{\ell}^{(1)} \right\} \right\} \sigma \left( V_{\ell}^{(1)} X_{\ell}^{(1)} \right)^+ \tag{45}$$

$\square$

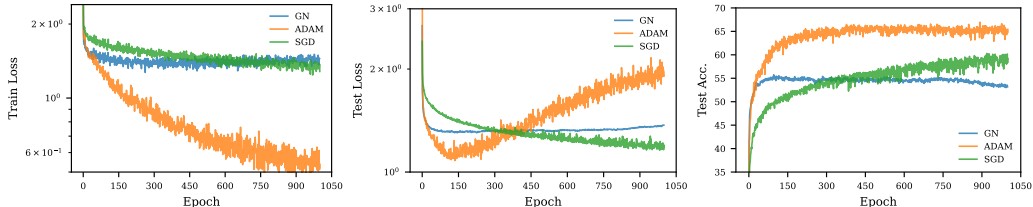

Figure 6: Training loss, test loss, and test accuracy when training for a longer amount of epochs on CIFAR-10 shows that Gauss-Newton is unable to further decrease the training loss, while even plain SGD can reach lower values.

## E  Longer training curves

Figure 6 displays the result of training the same RevMLP as in Section 5.2 for 1000 epochs on the CIFAR-10 dataset in a mini-batch setting ($n = 1024$). We observe that by continuing the training for longer on CIFAR-10, SGD is able to reach lower values of the training loss when compared to Gauss-Newton. In particular, we highlight that even in 1000 epochs, Gauss-Newton appears unable to increase its training performance further than the value it reaches after just 50 epochs. In fact, the results in Figure 6 show that Gauss-Newton tends to increase its training loss after 150 epochs of training.

## F  Initialization dependencies for Gauss-Newton

To examine if the poor performance of Gauss-Newton depends on a poor initialization point, we first train a model with Adam for 50 epochs, before continuing the training with Gauss-Newton. For comparison, we also train a model with Gauss-Newton for 50 epochs and subsequently continue training with Adam – to observe if Gauss-Newton reaches reaches a "bad" local minimum that is hard to escape from. These results are provided in Figure 7 and compared with their single optimizer counterparts. We choose a "good" initialization point as an Adam trained model at 50 epochs (indicated by the dashed line in Figure 7), which has a lower training loss than GN can achieve in the same (or larger) number of iterations. One can observe that, even when starting from this "good" initialization, GN eventually saturates at a higher value of the loss when compared with the values

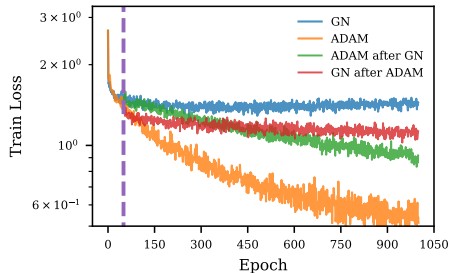

Figure 7: Training loss when first using Adam (or GN), and then continuing with GN (or Adam) – the purple dashed line indicates the 50 epochs mark at which the optimizers are switched. GN shows early saturation of the loss even when starting from a better intialization point.

achievable by continuing training with Adam. We also find that Adam can start from a point found by GN and continue training without issues, reaching a value of the loss that is lower than the saturation point of GN.

In reference to Arbel et al. [2023], we also provide additional results in Figure 8 to show the dependency of Gauss-Newton on the initial weight variance chosen. Interestingly, our results are different from those in Arbel et al. [2023] and suggest that choosing a higher variance is preferable. However, all curves exhibit the same phenomena as discussed in Section 5 and under-perform with respect to Adam. The default choice for all experiments we report is $\sigma = 10^{-3}$.

## G  Change in weights during training

We analyze the change in norm and cosine similarity between the weights at initialization and at the end of training when a model is trained with different optimizers. Results are shown in Figure 9. We

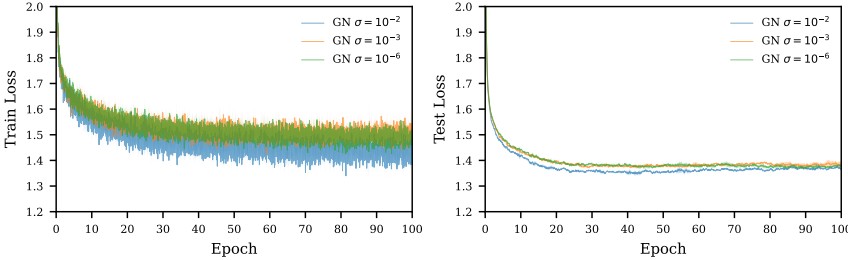

Figure 8: GN Train and test loss with weights initialized accounding to different variances at initialization $\sigma$.

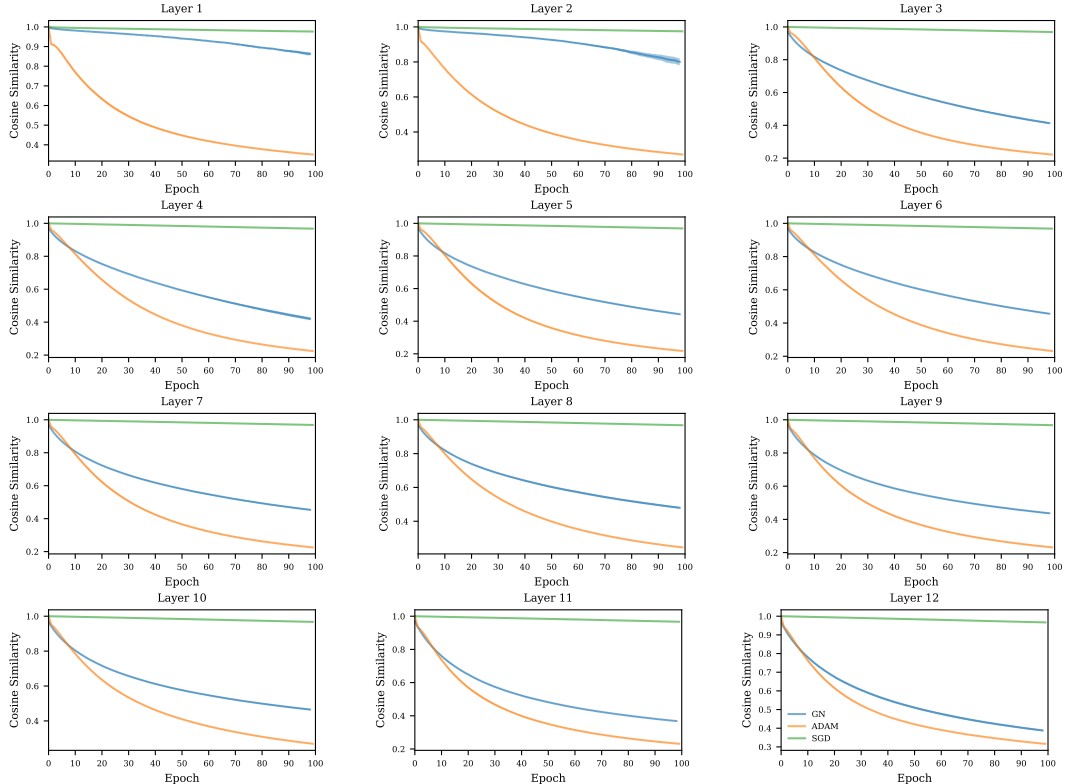

Figure 9: Cosine similarity with the initial weight initialization across training. ADAM and GN move similarly in weight space indicating a consistent behaviour in weight space between the two optimizers. Note that, in this Figure, we use the term "layer" to refer to half-coupled layer in the reversible blocks.

observe that Gauss-Newton changes the weights to an extent similar to Adam, while SGD show much smaller weight changes, suggesting a *lazy training* regime.

# H    Extended Centered Kernel Alignment (CKA) analysis

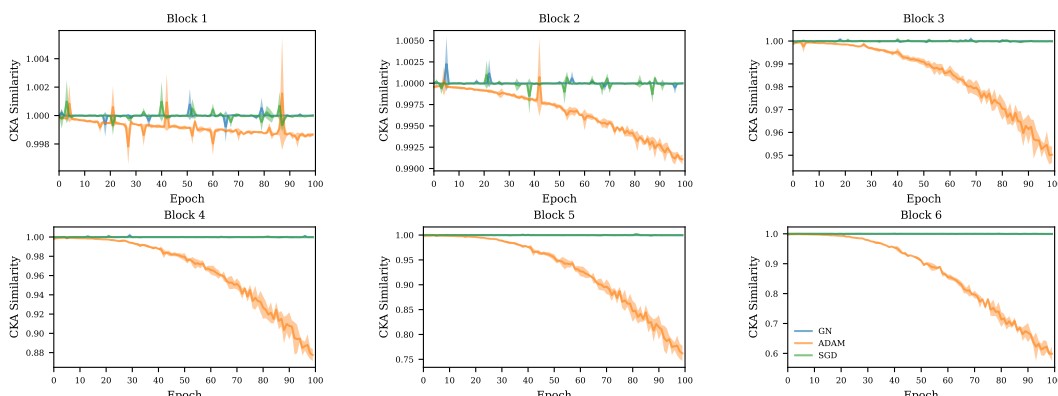

Figure 10: CKA similarity evolution across training for GN, Adam and SGD. GN maintains a high CKA similarity with its initial feature space, very similarly to SGD.

In Figure 10 we present the full spectrum of CKA similarities from initialization across blocks (i.e., full coupling layer) when training on CIFAR-10. We observe that GN behaves very much SGD: the representqations remain very similar to those at initialization. Adam instead tends to change the representations during training, more quickly for later blocks, and more slowly for earlier blocks.

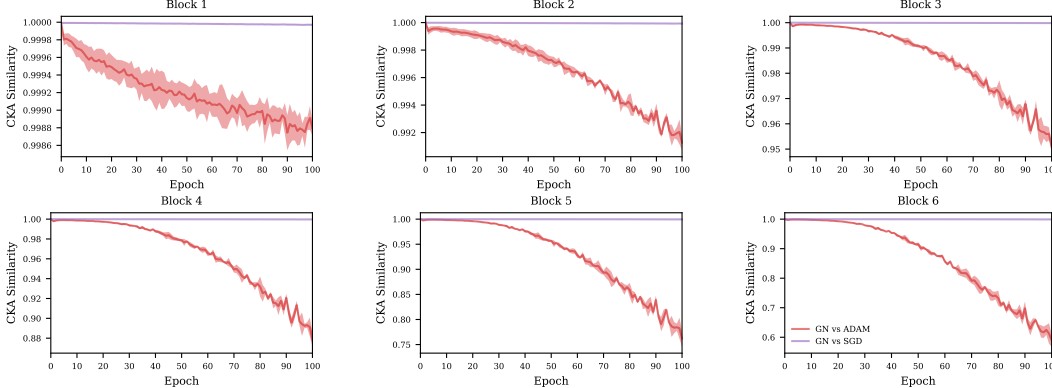

Figure 11: Pairwise CKA similarity evolution across training between GN and models trained with Adam and SGD.

In addition to the CKA evolution results for single optimizers, Figure 11 presents the pairwise similarities between models at each epoch trained with different optimization strategies. These results demonstrate explicitly the close correspondence between SGD and GN learned features for each block.

# I    Learning rate variations of Gauss-Newton

Figure 12 provides additional training runs of Gauss-Newton with different learning rates. These results indicate that forcing GN to learn slower is not sufficient to reduce the effect of the observed saturation of performance.

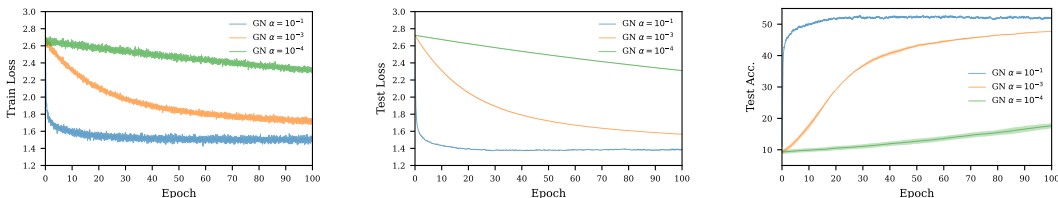

Figure 12: Train loss, test loss and test accuracy (left to right) for a RevMLP trained on CIFAR-10 with Gauss-Newton using different learning rates.

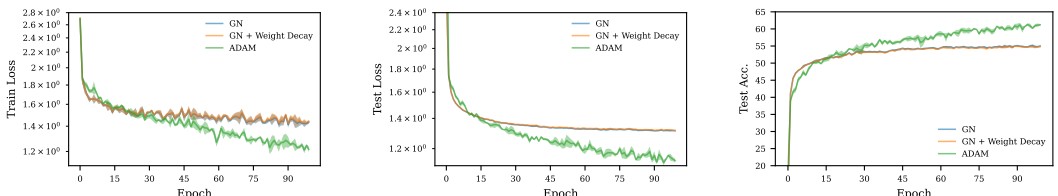

Figure 13: Train loss, test loss, and test accuracy (left to right) for a RevMLP trained on CIFAR-10 with Gauss-Newton with weight-decay. Adam is also added for comparison.

## J   Adding Regularization to Gauss-Newton

In this Section we explore wether weight-decay can be used to improve the performance of Gauss-Newton. We use a RevMLP on the full CIFAR-10 dataset with the same setting presented in Section 5 and we add weight decay to the loss during training. We tune the strength of the weight decay using the validation set. Results are shown in Figure 13. We notice that weight decay has a minimal effect of the performance of Gauss-Newton.

## K   Pseudo-Inverse Regularization

In this Section we explore the effects of different strategies for regularizing the pseudoinverse in the proposed Gauss-Newton update (see equations (16), (17)). We note that regularization is necessary, as the presence of very small singular values causes numerical instabilities. We compute the pseudoinverse using a singular value decomposition, and we try three different strategies:

- Damping: we add a constant to all the singular values. In particular we add a quantity equal to $1\%$ of the maximum singular value (this quantity of damping was tuned by selecting the best performing one over the values $1\%, 10\%, 0.1\%$).
- Truncation: we set to zero all the singular values smaller than a certain threshold. In particular, we use relative tolerance of $1\%$ with respect to the largest singular value and an absolute tolerance of $10^{-5}$ (we tune this values in a similar fashion to the previous method). This is the strategy used for the results in Section 5.
- Noise: we add noise to the matrix to be pseudoinverted; we then compute the SVD and use all singular values. The noise is sampled from a zero-mean Gaussian with a standard deviation equal to $10\%$ (this value was selected though a tuning procedure as above) of the standard deviation of the matrix to be pseudoinverted.

Results are shown in Figure 14, where Adam is also added for comparison. We notice that there is a small difference between damping and truncating (with the former performing slightly better), while adding noise does not seem as effective. Nevertheless, Gauss-Newton is always under-performing when compared to Adam.

## L   Full Hyperparameters & Experimental Details

In this Section we provide additional details on the hyperparameters and experimental details used for our experiments. Full code to reproduce our results is also provided with the submission.

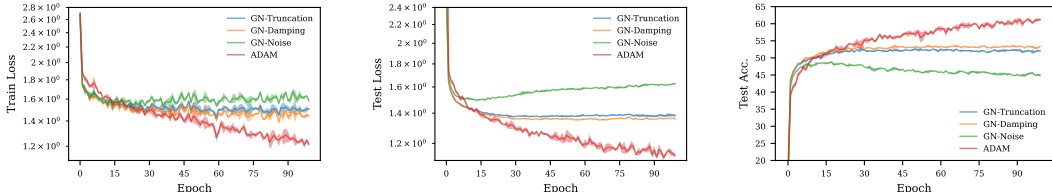

Figure 14: Train loss, test loss, and test accuracy (left to right) for a RevMLP trained on CIFAR-10 with Gauss-Newton using different regularization strategies for the pseudoinverse. Adam is also added for comparison.

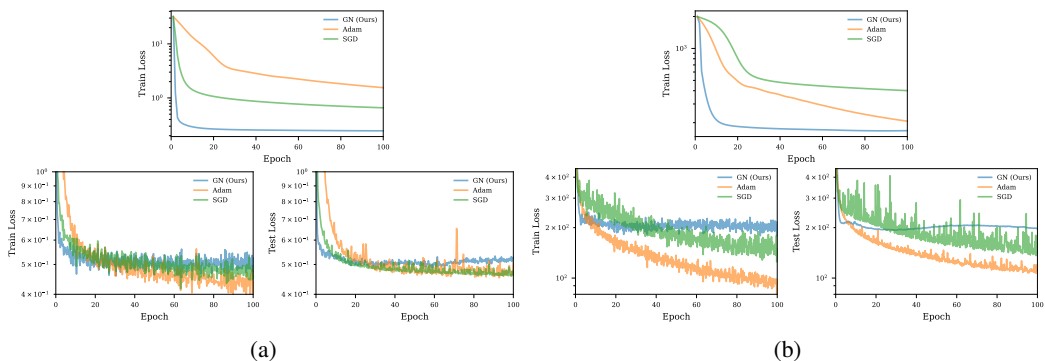

Figure 15: Train loss and test loss on UCI (a) wine and (b) superconductivity regression datasets. (Full-batch on top, mini-batch on bottom).

**Implementation details.** Our code is based on the PyTorch framework Paszke et al. [2019]. In more detail we use version 2.0 for Linux with CUDA 12.1.

**Weight initialization.** We use standard Xavier Glorot and Bengio [2010] initialization for the weights, while we initialize the biases to zero.

**Sampling the inverted bottleneck.** We sample the entries of each inverted bottleneck from a zero-centered gaussian with a variance of $\frac{1}{\text{layer dimension}}$.

**Data augmentations.** For the MNIST dataset we do not use any data augmentations. For the CIFAR-10 dataset we follow the standard practice of applying random crops and resizes. We do not use data augmentations for the regression datasets.

**Additional hyperparameters for Adam.** We tune the learning rate for each experiment and method, as explained in Section 5, and we use the PyTorch default values for the *betas* parameters in Adam.

# M   Regression Results

We report the results for the regression experiments in Figure 15.

# N  Proof Layer-wise Right-inverse is the Moore-Penrose Pseudo-inverse

Consider the Jacobian for layer $\ell$:

$$J_\ell = \begin{pmatrix} \frac{\partial x_L}{\partial x_\ell^{(1)}} & \frac{\partial x_L}{\partial x_\ell^{(2)}} \end{pmatrix} \begin{pmatrix} A & 0 \\ B & C \end{pmatrix} = \tag{46}$$

$$\begin{pmatrix} \frac{\partial x_L}{\partial x_\ell^{(1)}} & \frac{\partial x_L}{\partial x_\ell^{(2)}} \end{pmatrix} \begin{pmatrix} \sigma_\ell \left( V_{\ell-1}^{(2)} X_{\ell-1}^{(2)} \right)^T \otimes \mathrm{I}_{d/2} & \mathbf{0} \\ \frac{\partial x_\ell^{(2)}}{\partial w_\ell^{(1)}} & \sigma_\ell \left( V_\ell^{(1)} X_\ell^{(1)} \right)^T \otimes \mathrm{I}_{d/2} \end{pmatrix} \tag{47}$$

Denote $J_{\ell,1} = \begin{pmatrix} A & 0 \\ B & C \end{pmatrix}$. For our method, we used

$$J_{\ell,1}^{\dashv} = \begin{pmatrix} A^+ & 0 \\ -C^+ B A^+ & C^+ \end{pmatrix} \text{ with} \tag{48}$$

$$A^+ = \sigma_\ell \left( V_{\ell-1}^{(2)} X_{\ell-1}^{(2)} \right)^{T+} \otimes \mathrm{I}_{d/2}, \tag{49}$$

$$B = \frac{\partial x_\ell^{(2)}}{\partial w_\ell^{(1)}}, \tag{50}$$

$$C^+ = \sigma_\ell \left( V_\ell^{(1)} X_\ell^{(1)} \right)^{T+} \otimes \mathrm{I}_{d/2} \tag{51}$$

Then:

$$J_{\ell,1}^{\dashv} J_{\ell,1} = \begin{pmatrix} A^+ A & 0 \\ C^+ B - C^+ B A^+ A & C^+ C \end{pmatrix} \tag{52}$$

We show below that $B = B A^+ A$, and hence that $J_{\ell,1}^{\dashv} = J_{\ell,1}^+$ (i.e., our right-inverse corresponds to the Moore-Penrose Pseudo-inverse).

We can show that (see Section N.1):

$$B_{i+\frac{d}{2}(j-1),a+\frac{d}{2}(b-1)} = \left( \frac{\partial x_\ell^{(2)}}{\partial w_\ell^{(1)}} \right)_{i+\frac{d}{2}(j-1),a+\frac{d}{2}(b-1)} \tag{53}$$

$$= \sigma_\ell \left( V_{\ell-1}^{(2)} X_{\ell-1}^{(2)} \right)^T_{j,b} \sum_k (W_{\ell,G})_{i,k} \left( V_\ell^{(1)} \right)_{k,a} \sigma'_\ell \left( V_\ell^{(1)} X_\ell^{(1)} \right)_{k,j}. \tag{54}$$

With the above, we first compute $(BA^+)$:

$$(BA^+)_{i_1+\frac{d}{2}(j_1-1),i_4+\frac{d}{2}(j_4-1)} \tag{55}$$

$$= \sum_{i_2,j_2} (B)_{i_1+\frac{d}{2}(j_1-1),i_2+\frac{d}{2}(j_2-1)} (A^+)_{i_2+\frac{d}{2}(j_2-1),i_4+\frac{d}{2}(j_4-1)} \tag{56}$$

$$= \sum_{i_2,j_2} (B)_{i_1+\frac{d}{2}(j_1-1),i_2+\frac{d}{2}(j_2-1)} \sigma_\ell \left( V_{\ell-1}^{(2)} X_{\ell-1}^{(2)} \right)^{T+}_{j_2,j_4} \mathrm{I}(i_2 = i_4) \tag{57}$$

$$= \sum_{j_2,k} \sigma_\ell \left( V_{\ell-1}^{(2)} X_{\ell-1}^{(2)} \right)^T_{j_1,j_2} (W_{\ell,G})_{i_1,k} \left( V_\ell^{(1)} \right)_{k,i_4} \sigma'_\ell \left( V_\ell^{(1)} X_\ell^{(1)} \right)_{k,j_1} \sigma_\ell \left( V_{\ell-1}^{(2)} X_{\ell-1}^{(2)} \right)^{T+}_{j_2,j_4} \tag{58}$$

$$= \sum_k (W_{\ell,G})_{i_1,k} \left( V_\ell^{(1)} \right)_{k,i_4} \sigma'_\ell \left( V_\ell^{(1)} X_\ell^{(1)} \right)_{k,j_1} \cdot$$
$$\sum_{j_2} \left\{ \sigma_\ell \left( V_{\ell-1}^{(2)} X_{\ell-1}^{(2)} \right)^T_{j_1,j_2} \sigma_\ell \left( V_{\ell-1}^{(2)} X_{\ell-1}^{(2)} \right)^{T+}_{j_2,j_4} \right\} \tag{59}$$

$$= \sum_k (W_{\ell,G})_{i_1,k} \left( V_\ell^{(1)} \right)_{k,i_4} \sigma'_\ell \left( V_\ell^{(1)} X_\ell^{(1)} \right)_{k,j_1} \left( \sigma_\ell \left( V_{\ell-1}^{(2)} X_{\ell-1}^{(2)} \right)^T \sigma_\ell \left( V_{\ell-1}^{(2)} X_{\ell-1}^{(2)} \right)^{T+} \right)_{j_1,j_4} \tag{60}$$

and finally

$$(BA^+A)_{i_1+\frac{d}{2}(j_1-1),i_3+\frac{d}{2}(j_3-1)} \tag{61}$$

$$= \sum_{i_4,j_4}(BA^+)_{i_1+\frac{d}{2}(j_1-1),i_4+\frac{d}{2}(j_4-1)}A_{i_4+\frac{d}{2}(j_4-1),i_3+\frac{d}{2}(j_3-1)} \tag{62}$$

$$= \sum_{i_4,j_4}(BA^+)_{i_1+\frac{d}{2}(j_1-1),i_4+\frac{d}{2}(j_4-1)}\sigma_\ell\left(V_{\ell-1}^{(2)}X_{\ell-1}^{(2)}\right)^T_{j_4,j_3}\mathrm{I}(i_4=i_3) \tag{63}$$

$$= \sum_{j_4,k}(W_{\ell,G})_{i_1,k}\left(V_\ell^{(1)}\right)_{k,i_3}\sigma'_\ell\left(V_\ell^{(1)}X_\ell^{(1)}\right)_{k,j_1}\cdot$$
$$\left(\sigma_\ell\left(V_{\ell-1}^{(2)}X_{\ell-1}^{(2)}\right)^T\sigma_\ell\left(V_{\ell-1}^{(2)}X_{\ell-1}^{(2)}\right)^{T+}\right)_{j_1,j_4}\sigma_\ell\left(V_{\ell-1}^{(2)}X_{\ell-1}^{(2)}\right)^T_{j_4,j_3} \tag{64}$$

$$= \sum_{k}(W_{\ell,G})_{i_1,k}\left(V_\ell^{(1)}\right)_{k,i_3}\sigma'_\ell\left(V_\ell^{(1)}X_\ell^{(1)}\right)_{k,j_1}\cdot$$
$$\sum_{j_4}\left\{\left(\sigma_\ell\left(V_{\ell-1}^{(2)}X_{\ell-1}^{(2)}\right)^T\sigma_\ell\left(V_{\ell-1}^{(2)}X_{\ell-1}^{(2)}\right)^{T+}\right)_{j_1,j_4}\sigma_\ell\left(V_{\ell-1}^{(2)}X_{\ell-1}^{(2)}\right)^T_{j_4,j_3}\right\} \tag{65}$$

$$= \sum_{k}(W_{\ell,G})_{i_1,k}\left(V_\ell^{(1)}\right)_{k,i_3}\sigma'_\ell\left(V_\ell^{(1)}X_\ell^{(1)}\right)_{k,j_1}\cdot$$
$$\left(\sigma_\ell\left(V_{\ell-1}^{(2)}X_{\ell-1}^{(2)}\right)^T\sigma_\ell\left(V_{\ell-1}^{(2)}X_{\ell-1}^{(2)}\right)^{T+}\sigma_\ell\left(V_{\ell-1}^{(2)}X_{\ell-1}^{(2)}\right)^T\right)_{j_1,j_3} \tag{66}$$

$$= \sum_{k}(W_{\ell,G})_{i_1,k}\left(V_\ell^{(1)}\right)_{k,i_3}\sigma'_\ell\left(V_\ell^{(1)}X_\ell^{(1)}\right)_{k,j_1}\sigma_\ell\left(V_{\ell-1}^{(2)}X_{\ell-1}^{(2)}\right)^T_{j_1,j_3} \tag{67}$$

$$= B_{i_1+\frac{d}{2}(j_1-1),i_3+\frac{d}{2}(j_3-1)}. \tag{68}$$

## N.1 Expression for $B$

$$B_{i+\frac{d}{2}(j-1),a+\frac{d}{2}(b-1)} = \left(\frac{\partial x_\ell^{(2)}}{\partial w_\ell^{(1)}}\right)_{i+\frac{d}{2}(j-1),a+\frac{d}{2}(b-1)}$$

$$= \left(\frac{\partial(x_\ell^{(2)})_{i+\frac{d}{2}(j-1)}}{\partial(w_\ell^{(1)})_{a+\frac{d}{2}(b-1)}}\right)$$

$$= \left(\frac{\partial(X_\ell^{(2)})_{i,j}}{\partial(W_\ell^{(1)})_{a,b}}\right)$$

$$= \frac{\partial(W_{\ell,G}\sigma_\ell\left(V_\ell^{(1)}X_\ell^{(1)}\right))_{i,j}}{\partial(W_\ell^{(1)})_{a,b}}$$

$$= \sum_{k}(W_{\ell,G})_{i,k}\frac{\partial\sigma_\ell\left(V_\ell^{(1)}X_\ell^{(1)}\right)_{k,j}}{\partial(W_\ell^{(1)})_{a,b}}$$

$$= \sum_{k}(W_{\ell,G})_{i,k}\sigma'_\ell\left(V_\ell^{(1)}X_\ell^{(1)}\right)_{k,j}\frac{\partial\left(V_\ell^{(1)}X_\ell^{(1)}\right)_{k,j}}{\partial(W_\ell^{(1)})_{a,b}}$$

$$= \sum_{k,t}(W_{\ell,G})_{i,k}\sigma'_\ell\left(V_\ell^{(1)}X_\ell^{(1)}\right)_{k,j}\left(V_\ell^{(1)}\right)_{k,t}\frac{\partial\left(X_\ell^{(1)}\right)_{t,j}}{\partial(W_\ell^{(1)})_{a,b}}$$

$$= \sum_{k,t} (W_{\ell,G})_{i,k}\, \sigma'_\ell \left( V^{(1)}_\ell X^{(1)}_\ell \right)_{k,j} \left( V^{(1)}_\ell \right)_{k,t} \frac{\partial \left( W^{(1)}_\ell \sigma_\ell \left( V^{(2)}_{\ell-1} X^{(2)}_{\ell-1} \right) \right)_{t,j}}{\partial (W^{(1)}_\ell)_{a,b}}$$

$$= \sum_{k,t,s} (W_{\ell,G})_{i,k}\, \sigma'_\ell \left( V^{(1)}_\ell X^{(1)}_\ell \right)_{k,j} \left( V^{(1)}_\ell \right)_{k,t} \sigma_\ell \left( V^{(2)}_{\ell-1} X^{(2)}_{\ell-1} \right)_{s,j} \frac{\partial \left( W^{(1)}_\ell \right)_{t,s}}{\partial (W^{(1)}_\ell)_{a,b}}$$

$$= \sum_{k} (W_{\ell,G})_{i,k}\, \sigma'_\ell \left( V^{(1)}_\ell X^{(1)}_\ell \right)_{k,j} \left( V^{(1)}_\ell \right)_{k,a} \sigma_\ell \left( V^{(2)}_{\ell-1} X^{(2)}_{\ell-1} \right)_{b,j}$$

$$= \sigma_\ell \left( V^{(2)}_{\ell-1} X^{(2)}_{\ell-1} \right)^T_{j,b} \sum_{k} (W_{\ell,G})_{i,k} \left( V^{(1)}_\ell \right)_{k,a} \sigma'_\ell \left( V^{(1)}_\ell X^{(1)}_\ell \right)_{k,j}.$$

