# OpenReview forum: "Exact, Tractable Gauss-Newton Optimization in Deep Reversible Architectures Reveal Poor Generalization"
_NeurIPS.cc/2024/Conference — NeurIPS 2024 poster_

### Official Review · Reviewer_9txE · 2024-07-05

**Soundness:** 3
**Presentation:** 4
**Contribution:** 3
**Rating:** 7
**Confidence:** 3

**Summary:**

The paper address the problem of efficient computation of Gauss-Newton (GN) updates in deep neural networks and in particular the question whether GN results in better generalization behavior than SGD. For this, the authors devise a neural architecture based on reversible NNs that incorporates additional linear bottleneck layers and in which by application of Moore-Penrose pseudo-inverse the updates can be derived analytically and computed efficiently through Jacobian-vector products. Consequently, their analysis show that empirically GN struggles to learn useful representations on MNIST and CIFAR-10 with the selected model architecture and shows a significantly different feature learning behavior, i.e., they find a strong change in the NTK and high distances of the CKA wrt the initialization point, than SGD or Adam. The authors provide additional extensive ablations in the appendix and conclude that while GN yields fast convergence in the full batch setting it does not perform well in the stochastic setting in which it tends to overfit each individual batch rather than fitting to the data set.

**Strengths:**

The paper concisely presents analytic and efficiently computable GN updates for a class of reversible neural networks. The analysis is through and well executed and the results are interesting in my opinion.

**Weaknesses:**

The authors claim that their paper introduces exact updates for the first realistic application of neural networks. It is, however, unclear to me how realistic the constructed model is. Even though the invert bottleneck has been used in prior work (e.g., (Bachmann 2024)) these applications seem inflate the features only mildly while the authors seem to use a rather drastic inverse bottleneck to ensure linear independence. Hence, I am wondering how transferable the results actually are to architecture that are typically used and do not include random feature projections such as those used in the proposed work.

**Questions:**

1. Could the authors elaborate on how much the choice of the inverse bottleneck layer and its number of random weights effects the results.
2. From what I understand most of the derivation in the main text focuses on the squared loss. However, the results in the experimental section focus solely on the cross-entropy loss which would include an additional Hessian term. Could the authors elaborate on this and are there any comparable results for the setting of squared loss settings such as on UCI regression data sets?

**Limitations:**

The paper adequately addressed the limitations.

---

> ### Author Rebuttal · Authors · 2024-08-06
>
> We thank the reviewer for the thorough and detailed comments. We provide answers to questions and weaknesses below.
>
> > I am wondering how transferable the results actually are to architecture that are typically used and do not include random feature projections such as those used in the proposed work.
>
> This is a great question. As the reviewer noted, we used inverted bottlenecks to “ensure linear independence”, i.e. to ensure that the model is adequately overparameterized such that the efficient generalized inverse we propose is valid. In other words, inverted bottlenecks ensure that the scalable GN weight updates (Eqs. 16-17) do implement gradient flow in function space (Eq 3; the essence of the GN method), such that our results are not potentially confounded by broken theoretical assumptions. Nevertheless, we agree that our GN updates can still be applied in the absence of inverted bottlenecks: even though they may not enjoy the same theoretical guarantees, they could still lead to similar training behaviour and it is worth investigating this empirically. We did this on the CIFAR10 dataset, with results shown in Figure 2 of the one-page rebuttal PDF. We followed the same experimental procedure as in the paper, only removing all inverted bottlenecks, and we tuned the learning rate for each optimizer. In the full-batch setting, GN is still performing much better than Adam and SGD. In the mini-batch setting we observe a very similar trend to what is shown in our main paper: GN leads to an early saturation of the loss, which instead does not appear in Adam and SGD. We thank the reviewer for this suggestion, and we will include these results in the final version of the paper.
>
> > [...] are there any comparable results for the setting of squared loss settings such as on UCI regression data sets?
>
> We thank the reviewer for the valuable suggestion to apply our method directly to regression tasks with a squared loss. We have added some new experiments on two of the UCI Regression datasets (Wine Quality & Superconductivity), with results shown in Figure 4 of the one-page rebuttal PDF. These results corroborate our main findings in the cross-entropy / classification setting: in the full-batch case GN is significantly faster than SGD and Adam, while in the mini-batch case  there is an apparent stagnation of the test and train losses under GN.
>
> We hope that the additional analyses address the reviewer's concerns and that they may consider raising the score of our paper.

---

> > ### Comment · Reviewer_9txE · 2024-08-09
> > **Response**
> >
> > I have read the authors responses and reviews of fellow reviewers. I am very pleased about the rebuttal and will be happy to increase my score.
> >
> > I do not have additional questions to the authors.

---

> ### Author Response · Authors · 2024-08-09
> **Re: Response**
>
> We thank the reviewer for the quick reply. We are glad our rebuttal could help clarify doubts, and that the reviewer is happy to raise the score. We however notice that the previous score of 6 has not been updated, so we kindly ask the reviewer to edit the score.

---

> > ### Comment · Reviewer_9txE · 2024-08-09
> >
> > I’ll update the score at the end of the discussion phase to be able to accommodate for discussion.

---

> > > ### Comment · Reviewer_9txE · 2024-08-12
> > >
> > > I thank the authors again for their responses and have adjusted my score to an accept.

---

### Official Review · Reviewer_TbBj · 2024-07-12

**Soundness:** 4
**Presentation:** 4
**Contribution:** 3
**Rating:** 6
**Confidence:** 5

**Summary:**

In this work the authors use reversible neural networks to explore the benefits of exact Gauss-Newton optimization. They provide a theoretical framework for efficient Jacobian pseudoinverses in reversible networks. The authors then provide experiments on MNIST and CIFAR10 comparing SGD, ADAM, and SGD-GN training. They find that on a small (1024) subset of the dataset, full batch training, GN performs well; in the minibatch setting however GN training performs very poorly. In this setting SGD performs poorly in general. They also measure different feature learning metrics and show that GN training performs feature learning after most of the optimization has occured.

**Strengths:**

The paper provides a very clean explanation of Gauss Newton training, and writing it down in terms of the pseudoinverse of the Jacobian is also a nice touch. The use of the reversible neural networks is very clever, and allow for study of the exact dynamics of interest rather than mere approximations. The experiments are basic but very clear, and the paper overall provides some intriguing preliminary results as well as a path towards future studies into GN training.

**Weaknesses:**

The experiments section could be more full; for example, exploring the batch size dependence more fully (or, extending the full batch examples to more datapoints). I would also be interested to see how GN training performs in a setting where SGD works at least as well as ADAM.

In addition, it would be helpful if more intuitions about the form of the pseudoinverse of J are brought into the main text.

**Questions:**

What happens in the full batch experiments with batch size 2048?

What happens if L2 regularization is added to the experiments?

---

> ### Author Rebuttal · Authors · 2024-08-06
>
> We thank the reviewer for their positive appraisal of our work, and for the constructive feedback. Below, we provide answers to their questions and address the weaknesses they have identified.
>
> > The experiments section could be more full; for example, exploring the batch size dependence more fully (or, extending the full batch examples to more datapoints). I would also be interested to see how GN training performs in a setting where SGD works at least as well as ADAM.
>
> We followed the reviewer's suggestion to extend the full-batch example to 2048 samples, which can be found in Figure 1 in the one-page supplemental PDF. We used the same architecture and experimental procedure as in the main paper, including tuning the learning rate for each optimizer. The overall trend remains unaltered: in the full-batch setting, GN is significantly faster than both Adam and SGD; in the mini-batch setting, GN decreases the training and test loss faster initially, but then causes them to saturate early, while they continue to decrease for Adam (these effects on train/test losses are also reflected in classication accuracies). In general, our experiments have not revealed any important difference in behaviour when changing the batch size. We thank the reviewer for highlighting this aspect, and we will include this new result in the final version of the paper.
>
> The reviewer will also be interested in our response to Reviewer 9txE, where we show results of additional experiments on two of the UCI regression datasets (Wine Quality and Superconductivity). For these experiments, we used a squared error loss to complement our other classification experiments which used a cross-entropy loss. The results are shown in Figure 4 of our one-page supplemental PDF, and they do corroborate our main conclusions. Interestingly, Adam and SGD perform similarly on the Wine Quality dataset in the mini-batch setting -- a situation which the reviewer wished we had explored --, and even in this setting, GN exhibits similar overfitting characteristics to those reported in our original submission.
>
> > What happens in L2 regularization is added to the experiments?
>
> That is a great question -- L2 regularization being a very standard way of mitigating overfitting, it had also occurred to us that it could be important in addressing the overfitting behaviour of GN that we describe.  Experiments with L2 regularization can be found in Appendix K of the main submission; following the standard prescription of AdamW (Loshchilov & Hutter, 2017) we implemented L2 regularization as weight decay in all three optimizers. We found that adding L2 regularization has almost no effect on the overfitting behaviour of GN (we have tuned the amount of weight decay over 5 runs with different order of magnitudes). We also tried additional forms of regularization in Appendix L and came to similar conclusions.
>
> > It would be helpful if more intuitions about the form of the pseudoinverse of J are brought into the main text
>
> We agree that providing higher-level intuitions about the form of our generalized inverse of J would be useful. We had attempted some of that already through our word-only description of the pseudoinversion on lines 234-240, but we will rewrite this to be clearer and more specific. In addition to that, we will mention that, in the overparameterized setting, the Moore-Penrose pseudo-inverse would determine a parameter update with minimum Euclidean norm, among those providing descent in function space. Instead, the update given by our right inverse does not have minimum Euclidean norm, but it may minimize a different type of norm that is currently unknown and will be the subject of future studies.
>
> We hope that the additional analyses address the reviewer's concerns and that they may consider raising the score of our paper.

---

> > ### Comment · Reviewer_TbBj · 2024-08-08
> > **Response to rebuttal**
> >
> > Thanks to the authors for their response; I appreciate the additional experiments. I will maintain my review score at this time.

---

### Official Review · Reviewer_5mk4 · 2024-07-14

**Soundness:** 2
**Presentation:** 3
**Contribution:** 3
**Rating:** 5
**Confidence:** 4

**Summary:**

Even though the Gauss-Newton method is known as an effective second-order optimization, it suffers from intractability of Jacobian pseudoinverse computation.
This paper proposes a fast and efficient optimization method which solves the intractability issue of the Jacobian pseudoinverse in Gauss-Newton optimization method in overparameterized neural networks.
First, the Gauss-Newton optimization is re-interpreted as the functional view, corresponding to the gradient descent in a function space, taking the parameters as input.
Then, with this perspective, the pseudoinverse can be replaced to the generalized inverse matrix, which yields the equivalent convergence properties, by applying the chain rule of the loss gradient into the functional loss gradient times the parameter derivative, and replace the functional loss gradient into the same form using the generalized inverse.
Then, with this, the newly proposed exact Gauss-Newton method calculates the right inverse matrix of the Jacobian, with the RevMLP architecture.

And the authors showed that this newly proposed Gauss-Newton method, however, deploys an overfitting property, compared to the Adam and SGD methods, showing worse test accuracy compared to the Adam and SGD optimization methods. Finally, this paper suggests several hypothesis, such as the minibatch overfitting and the feature learning on the NTK regime.

**Strengths:**

* Even though this paper contains heavy mathematical details, this paper is clear to follow.
* The proposed method enabled Gauss-Newton optimization method to be tractable in reversible nonlinear models.
* With experiments, this showed that even though the final result did not achieve test-set improvements, the initial learning curve is much more steeply learned: this implies that the Gauss-Newton method is well applied.

**Weaknesses:**

* The method is limited to reversible nonlinear models, which is a strongly restricted class of neural networks. Because of this restriction, the prediction performance is worse than conventional results.
* The gain of replacing pseudoinverse to a generalized inverse is not clear.

For further questions, please refer to Questions.

**Questions:**

* Is assuming $\texttt{RevMLP}$, a reversible neural network, setting feasible to use? It seems that the accuracy with CIFAR-10 dataset has a scale of 70%, which is quite lower than our consensus.
* I'd like to confirm that the reason of overfitting with respect to each minibatch. What happens if the minibatches are re-shuffled after each iteration? When the test set performance is saturated to minibatches, it will take advantage of the randomness of minibatches when shuffling.
* With the CKA results, it is expected that the Gauss-Newton learning schemes, which becomes enabled by the reversibility of the model, is more effective when the model gets deeper. Then, what happens with the performance when the reversible model gets deeper or shallower?
* How is the $V_{\ell}$ defined? I assumed that it is a class of random matrices, such as random Gaussian or random Fourier transform matrix.

====

* (Line 63) batchatches $\to$ batches
* (Line 238) reversed $\to$ reversible?

---

> ### Author Rebuttal · Authors · 2024-08-06
>
> We thank the reviewer for their time reviewing our paper -- we are glad they found it clear to follow.
>
> The reviewer wondered what is gained by replacing the Moore-Penrose pseudoinverse by a generalized inverse. To the best of our knowledge, there is no known tractable way of computing the Moore-Penrose pseudoinverse for deep nonlinear networks. This not only precludes its use in applications, but also makes it difficult to investigate the properties of GN at scale. Thus, the main “gain” of our generalized inverse is that it offers a computationally tractable expression for a weight update that implements gradient flow in function space (the core principle underlying the GN method, as summarized around Eq. 4).
>
> The reviewer rightly noted that our generalized inverse applies to reversible network archictures only, which we understand can appear restrictive. Coupling layers (Dinh et al, 2015; the form of reversible blocks that we use here) were originally introduced as a technical work-around for efficient computation of Jacobians and inverses (the same reason we use them here). They have since been very influential in the area of generative modelling (normalizing flows, diffusion models, ...) where they achieve SOTA results, and in other settings for learning flexible isomorphisms. The reversible vision transformer (Mangalam et al., 2022) also achieves near-SOTA results, and Liao et al (NeurIPS 2024) have shown that even large pre-trained LLMs can be made reversible by inserting specific adaptors, leading to memory-efficient fine-tuning. In fact, we believe that extending our efficient GN updates to these other classes of models, in addition to developing new theory to address the overfitting behaviour we have uncovered, is a promising future direction.
>
> > With the CKA results, it is expected that the Gauss-Newton learning schemes, which becomes enabled by the reversibility of the model, is more effective when the model gets deeper. Then, what happens with the performance when the reversible model gets deeper or shallower?
>
> As the model grows deeper, we find no significant performance increase for GN training, whereas Adam and SGD improve by a small amount towards convergence. As the model gets shallower, we find that performance drops in all methods by a similar amount.
>
> The reviewer's mention of CKA analysis in this context prompted us to think about alternative ways of studying the emergence of representations in RevMLPs. In the paper as it currently stands, our CKA analysis is performed separately for each half-coupling layer (i.e. separately for Eq. 14 and Eq. 15). We wondered what the CKA similarity would look like at the level of the 'full' coupling layers (i.e. the hidden representation jointly contained in the concatenation of $x_\ell^{(1)} $ and  $ x_\ell^{(2)} $ for each block $\ell$). We computed these CKA similarities (w.r.t. before-training representations at initialization) and found that they remain very high throughout training (Figure 3 of one-page rebuttal PDF). These results show that the large (albeit late) changes in half-coupling-layer representations observed in our original analysis were in fact _coordinated_ between the two half layers in each block, such that the compound representation which they jointly hold does not actually change. This finding is of course completely in line with our main conclusions, and to some degrees even simplify the story (we no longer need to appeal to the fact that the observed changes in (half-layer) representations happen _after_ GN has already reached its final performance). We are grateful to the reviewer for prompting us to think along those lines.
>
> > Is assuming RevMLP, a reversible neural network, setting feasible to use? It seems that the accuracy with CIFAR-10 dataset has a scale of 70%, which is quite lower than our consensus.
>
> The accuracy we obtained using RevMLPs on CIFAR10 is in fact on par with classical MLPs (~60-70\% test accuracy without data augmentation). MLPs, whether reversible or not, are just very prone to overfitting on this type of vision problems. Prior to submission, we did experiment with our own reversible version of MLP-Mixer (Tolstikhin et al, 2021; essentially using RevMLPs instead of MLPs in each mixer block), for which we were able to derive an analogous Jacobian generalized inverse. For these rev-MLP-Mixers, performance on CIFAR10 was overall much better ($> 90\%$ test accuracy with Adam), similar to what was reported in the non-reversible version, showing that the specific reversibility property does not really affect performance. Nevertheless, we had found that GN displayed identical overfitting properties in rev-MLP-Mixer as it did in plain RevMLPs, but training runs took much longer. For this reason we decided to simplify both the exposition and our experiments by focusing our paper on plain RevMLPs.
> We will add a paragraph of discussion on this in our revision.
>
> > I'd like to confirm that the reason of overfitting with respect to each minibatch. What happens if the minibatches are re-shuffled after each iteration?
>
> In our experiments in the minibatch setting, we can confirm that the entire dataset is re-shuffled at the beginning of each epoch before getting divided into minibatches; thus, each minibatch is (statistically) unique; what we show is that GN tends to overfit to each minibatch, and this isn't rescued by learning slowly over many randomized minibatches.
>
> > How is the V_t defined? I assumed that it is a class of random matrices, such as random Gaussian or random Fourier transform matrix.
>
> Apologies for this omission; we have now added a full description of how these inverted bottleneck weights are drawn: i.i.d. from a normal distribution (as the reviewer suspected) with zero mean and variance $2/d$ (the input to those have dimensions $d/2$, hence the factor of 2).

---

> > ### Comment · Reviewer_5mk4 · 2024-08-14
> > **Response**
> >
> > Thank you for the detailed and response.
> >
> > > __Question__\
> > > Importance of Reversible Networks
> >
> > My largest concern was that this method is limited to reversible networks. If using the reversible architecture is not useful, then this method would not be feasibly working. However, the authors seem to successfully argue the importance of the reversible networks.
> >
> > > CKA Analysis
> >
> > The CKA analysis in the full coupling layer, merging (14) and (15), the new findings on merging two layers on CKA analysis makes sense. I further understand the analysis and findings after the attached pdf, and recommend to include these figures into the main text.
> >
> > ---
> >
> > My largest concerns were (1) usefulness and feasibility of reversible network, and (2) correspondence between the analysis and conclusion, and both seem to be more resolved. So I adjust my rating.

---

> ### Author Response · Authors · 2024-08-13
> **End of discussion period**
>
> We kindly notify the reviewer that the discussion period is reaching an end. We believe we have addressed all the raised concerns, which has also helped us improve the paper, and we would be happy to receive feedback and answer any remaining doubt

---

### Author Rebuttal · Authors · 2024-08-06

We thank all reviewers for their time reviewing our paper; we are glad the reviewers appreciated our paper's main strengths, found it “clear to follow” with a “very clean explanation of Gauss Newton” and “thorough and well executed” analysis, opening “a path towards future studies into GN training”. We also thank the reviewers for raising interesting questions and making valuable suggestions for additional experiments, which we have done for the most part.

In particular, in the rebuttal PDF, we provide the following new results:
- Mini-batch and full-batch results on CIFAR with a batch size of 2048
- Mini-batch and full-batch results on CIFAR without inverted bottleneck
- CKA results considering the output of each layer (and not of each component of the coupling layer as done in the paper)
- Mini-batch and full-batch results on two regression datasets from the UCI library.

We provide reviewer-specific responses below.

In addition to these, we would like to bring to the reviewers' attention a minor error in our derivation of Proposition 4.4, which we have now corrected and which did not affect our main conclusions. In the original derivation, we had inadvertently omitted a term that arises from the relationship between the two half-coupled layers inside each coupling block. Correcting for this, Proposition 4.4 becomes:

Proposition 4.4: Assuming $\sigma(V_{\ell-1}^{(2)} X_{\ell - 1}^{(2)})$ and $\sigma(V_\ell^{(1)} X_\ell^{(1)})$ have linearly independent columns,


$W_\ell^{(1)}(t+1)= W_\ell^{(1)}(t)- \frac{\alpha}{L}
 \mathcal{R}^{(\frac{d}2, n)} \left[(\partial\mathbf{x}^1_\ell / \partial x_L) \boldsymbol\epsilon \right]
    \sigma\left(V_{\ell-1}^{(2)} {X_{\ell - 1}^{(2)}}\right)^+ = W_\ell^{(1)}(t)- \frac{\alpha}{L} \Delta^{(1)}$

$W_\ell^{(2)}(t+1) = W_\ell^{(2)}(t) - \frac{\alpha}{L} \mathcal{R}^{(\frac{d}2, n)}
[(\partial \mathbf{x}^2_{\ell} / \partial x_L) \boldsymbol\epsilon - (\partial \mathbf{x}^2_{\ell} / \partial  \mathbf{w}_{\ell}^{(1)}) \mathcal{R}^{(\frac{d}2, d')^{-1}}  \Delta^{(1)}]$

$\sigma \left(V_{\ell}^{(1)} X_\ell^{(1)} \right)^+$ (this term should be at the end of the above equation but openreview has issues rendering it)

We have re-run all our experiments with this modification (which we had spotted shortly after submission) and the new loss curves are almost identical to the old ones; nevertheless we thought we ought to let you know of this change.

---

### Decision · Program_Chairs · 2024-09-25

**Decision:**

Accept (poster)

**Comment:**

The use of second order methods such as Gauss Newton for neural network training has recently received renewed attention. This paper makes two distinct contributions to the study of this class of methods. To enable the study of large-scale Gauss-Newton training without the additional error due to the use of tractable Hessian approximations (like, say, Kronecker approximations), the authors develop a new approach that enables the efficient computation of the exact Hessian pseudoinverses in the special setting of invertible neural networks.
They then use this approach to thoroughly study the training behavior of exact Gauss Newton in this setting.

The reviewers appreciated the cleverness of the invertible Hessian trick and the thoroughness of numerical studies. The main perceived limitation of this work was whether the reversible structure is sufficiently representative for machine learning problems of interest. In other words, the authors found an alternative structure to rule out artifacts due to Hessian approximation, but to what extent are their results artifacts of the nonstandard architecture enabling this?

Nevertheless, there was consensus among the reviewers that this work is worth presenting at NeurIPS, which is why I recommend acceptance as a poster.